# A stress-activated neuronal ensemble in the supramammillary nucleus produces anxiety-like behavior in male mice

Jinming Zhang[1], Kexin Yu[1], Junmin Zhang[1], Yuan Chang[2], Xiao Sun[1], Zhaoqiang Qian[3], Zongpeng Sun[4], Yanning Qiao[1], Zhiqiang Liu[1], Wei Ren[1,5], Jing Han[1]*

[1]Key Laboratory of Modern Teaching Technology, Ministry of Education, Shaanxi Normal University, Xi'an, China; [2]Department of Histology and Embryology, School of Basic Medical Science, Xi'an Medical University, Xi'an, China; [3]Laboratory Animal Center, Shaanxi Normal University, Xi'an, China; [4]School of Psychology, Shaanxi Normal University, Xi'an, China; [5]Faculty of Education, Shaanxi Normal University, Xi'an, China

*For correspondence: jhan2012@snnu.edu.cn

Competing interest: The authors declare that no competing interests exist.

## eLife Assessment

This manuscript provides a **valuable** contribution by identifying a stress-responsive circuit and its regulation of anxiety-related behaviors. The evidence is **convincing** that the supramammillary nucleus contains stress-responsive neurons that increase anxiety-like behaviors when activated, and that ventral subiculum projections to the supramammillary are also activated by stress and their inhibition alleviates some effects of stress. Evidence that this pathway encodes and is functionally specific to anxiety is, at present, not sufficiently support and will require future studies. This work offers new insights into how distinct circuits are activated by stress and can regulate emotional behaviors and will be of interest to those interested in brain systems of aversive emotional and behavioral states.

**Abstract** Anxiety is a prevalent negative emotional state induced by stress; however, the neural mechanism underlying anxiety is still largely unknown. We used acute and chronic stress to induce anxiety and test anxiety-like behavior; immunostaining, multichannel extracellular electrophysiological recording, and $Ca^{2+}$ imaging to evaluate neuronal activity; and virus-based neuronal tracing to label circuits and manipulate circuitry activity. Here, we identified a hypothalamic region, the supramammillary nucleus (SuM), that plays an important role in anxiety-like behavior. We then characterized a small ensemble of stress-activated neurons (SANs) that are recruited by stress. These SANs respond specifically to stress, and their activation robustly increases anxiety-like behavior in male mice. We also found that ventral subiculum (vSub)-SuM projections, but not dorsal subiculum (dSub)-SuM projections, encode anxiety-like behavior and that inhibition of these vSub-SuM projections has an antianxiety effect. These results indicate that the reactivation of stress-activated supramammillary cells and relevant neural circuits is an important neural process underlying anxiety-like behavior.

## Introduction

Anxiety is a fundamental negative emotion observed in almost all mammal species. Long-lasting and uncontrollable anxiety often leads to several mental disorders, anxiety disorders, and even depression

(*Kesner et al., 2021*). Recent studies have shown that the supramammillary nucleus (SuM), a part of the hypothalamus, regulates sleep (*Qin et al., 2022*), memory (*Li et al., 2022b*; *Li et al., 2022a*), novelty exploration (*Li et al., 2022b*; *Pedersen et al., 2017*), social memory (*Farrell et al., 2021*; *Pan and McNaughton, 2002*), neurogenesis (*López-Ferreras et al., 2020*), consciousness (*López-Ferreras et al., 2019*; *Liang et al., 2023*), locomotor activity (*Tonegawa et al., 2015*), and theta oscillations in the hippocampus (*Josselyn and Tonegawa, 2020*). Projections from the SuM to the hippocampus have been largely studied and were found to modulate either episodic memory (*Li et al., 2022b*) or social memory (*Farrell et al., 2021*; *Pan and McNaughton, 2002*) depending on the subregion of the hippocampus targeted (*Chen et al., 2020*). Although the SuM is located near the mammillary nucleus, a key region implicated in emotion regulation via the Papez circuit, its role in regulating emotion has been explored only superficially, without in-depth investigation. Despite some discussions of this role of the SuM, no consistent conclusion has been reached thus far (*López-Ferreras et al., 2020*; *Liu et al., 2012*; *Guenthner et al., 2013*).

Activity-dependent activation of cells has been studied in many brain areas (*Sun et al., 2023*; *Azevedo et al., 2019*). Tagged cells react to specific stimuli, such as conditional stimuli (*Koren et al., 2021*; *Ryan et al., 2015*), pain (*Yan et al., 2022*), food (*Jimenez et al., 2018*), and even peripheral inflammation (*Forro et al., 2022*), and mediate the storage and retrieval of relevant memories. The manipulation of those believed memory-associated cells can alleviate neurodegenerative diseases (*Strange et al., 2014*) or inflammation (*Forro et al., 2022*). Naturally, this has led us to consider whether there is a special neuronal ensemble that plays roles in regulating anxiety or anxiety-like behaviors. Recent studies have focused on the role of the hippocampus and related neuronal afferents and efferents (*Kesner et al., 2023*; *Cumbers et al., 2007*; *Silveira et al., 1993*). The dorsal part of the hippocampus mainly contributes to cognition, whereas the ventral hippocampus is often associated with emotion (*Escobedo et al., 2023*; *LeDuke et al., 2023*). Although the ventral hippocampus-hypothalamus circuit was reported to modulate anxiety (*Kesner et al., 2023*; *Cumbers et al., 2007*), it is still unknown whether the SuM is part of this regulatory circuit. Although the SuM sends and receives dense neuronal projections, few studies have focused on its afferents or its ability to modulate behavior and emotion (*Aranda et al., 2006*).

In this study, we hypothesize that stress can recruit a special neuronal ensemble that exclusively encodes anxiety. To test this hypothesis, we first used multiple methods to assess whether the SuM responds to acute or chronic stress. The activity of the SuM was chemogenetically manipulated, and anxiety-like behavior in rodents was tested. We subsequently employed the targeted recombination in active populations (TRAP) strategy to label and manipulate stress-activated neurons (SANs) in SuM in terms of anxiety-like behavior. After demonstrating how the SuM modulates anxiety, we sought to identify upstream brain areas that may contribute to supramammillary function. We also examined the functions of neuronal projections from the ventral subiculum (vSub) to the SuM using fiber photometry to measure calcium dynamics, as well as chemogenetic manipulation. These results allowed us to characterize a previously unreported role of SuM in regulating anxiety-like behavior. In addition, we showed that projections from the vSub, but not the dorsal subiculum (dSub), to the SuM govern chronic stress-induced anxiety-like behaviors.

## Results

### Stress increases neuronal activity in the SuM

c-Fos protein expression was assessed after acute stress exposure to test whether the SuM was activated (*Figure 1A*). The number of c-Fos$^+$ cells was significantly increased by foot shock exposure (*Figure 1B and C*). To investigate if SuM would be responsive to diverse stressors, we next examined whether chronic stress, which has different mechanisms underlying, affects neuronal activity in the SuM (*Figure 1D and E*). We chose in vivo electrophysiological extracellular recordings to reveal the neuronal activity before and after chronic social defeat stress. The data shows that the firing rate of regular-spiking neurons (RNs) (*Figure 1F*) but not fast-spiking neurons (FNs) (*Figure 1—figure supplement 1A and B*) increased after CSDS. Regarding local field potentials, there were no noticeable differences between the naïve and CSDS groups according to power spectrum analysis (*Figure 1—figure supplement 1C and D*). These results indicate that acute and chronic stress can strongly activate the SuM.

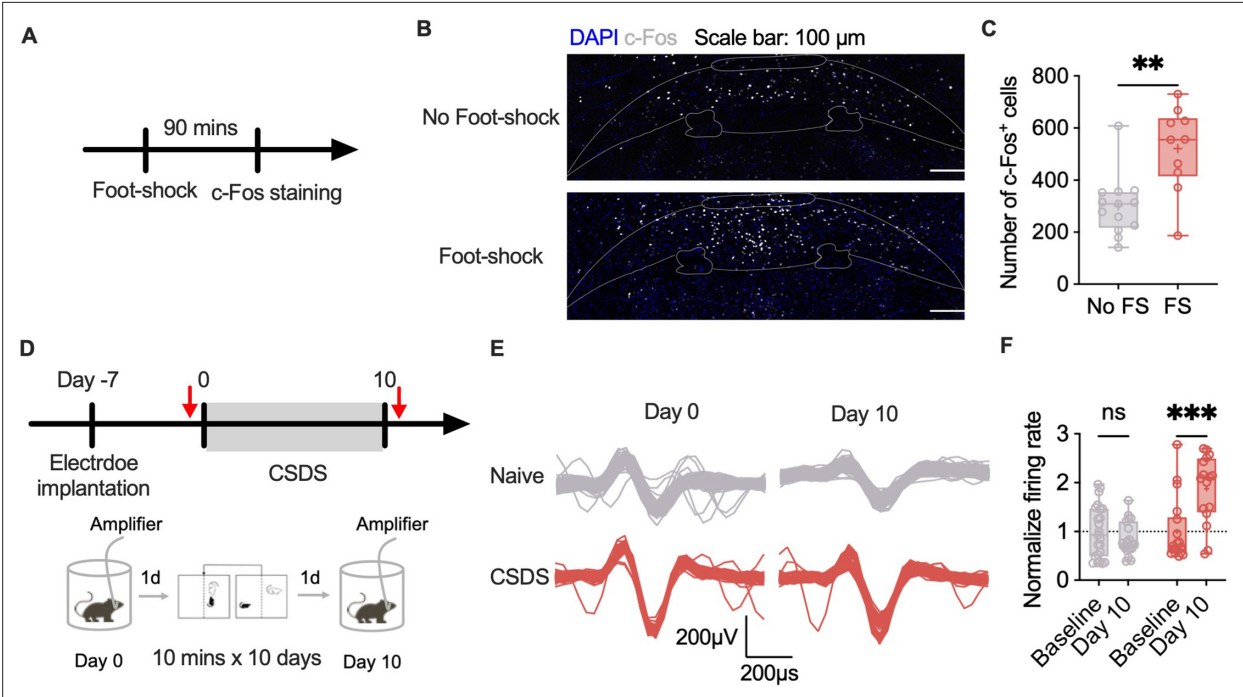

**Figure 1.** Stress activates the supramammillary nucleus (SuM). (**A**) Workflow of the c-Fos staining. (**B**) Representative images of c-Fos staining (DAPI: blue; c-Fos: white; scale bar: 100 μm). (**C**) Statistical analysis of the number of c-Fos-positive cells displayed in panel B. n=10–13 per group; unpaired t test. (**D**) Workflow of CSDS exposure and in vivo recording. (**E**) Representative spikes acquired by multichannel recording. (**F**) Statistical analysis of the firing rates of regular-spiking neurons (RNs) at baseline and after CSDS exposure. n=16–23 per group; two-way ANOVA followed by Sidak's post hoc test. The bars in C and F indicate the Min to Max of all data points, and the '+' indicates mean value of all data points. '**', p<0.01; '***', p<0.001. CSDS: chronic social stress; FS: foot shock.

The online version of this article includes the following source data and figure supplement(s) for figure 1:

**Source data 1.** The number of c-Fos+ cells in panel C and normalized firing rate in panel F, and corresponding statistical results.

**Figure supplement 1.** The effect of CSDS on the local field potential (LFP) in the supramammillary nucleus (SuM).

## Activation of SuM produces anxiety-like behavior

After confirming the activation of SuM caused different types of stress, we then further investigate whether the SuM regulates anxiety behavior in mice. Chemogenetic manipulations were conducted to activate SuM neurons. The experiments were performed as shown in the workflow (*Figure 2A and B*). The mice were subjected to the open field (OF) and elevated zero maze (EZM) tests at least 2 weeks after virus injection, followed by a reward-seeking test (*Figure 2E–H*, *Figure 2—figure supplement 1A*). Clozapine N-oxide (CNO) was administered intraperitoneally 30 min before the test. Chemogenetic activation of the SuM did not affect the performance of mice in the OF test (*Figure 2E and F*, *Figure 2—figure supplement 1B*). Compared with control mice, mice in which the SuM was activated explored the open arms of the EZM less (*Figure 2G*), despite no change in distance traveled (*Figure 2—figure supplement 1C*). Moreover, mice in which the SuM was activated consumed less food than control mice did (*Figure 2F*). These data suggest that there are neuronal ensembles that control the expression of anxiety behavior.

## Identification of SANs in the SuM

We next investigated whether an ensemble that encodes stress and controls the expression of anxiety exists. By crossbreeding Fos 2A-iCreERT2(TRAP2) and Rosa26-LSL-tdTomato (Ai14) mice, we generated TRAP2;Ai14 mice in which activated cells were genetically tagged for visualization (*Figure 3A*). Foot shock exposure strongly activated neurons in the SuM but not adjacent areas (*Figure 3B and C*). We hypothesized that these SANs respond exclusively to stress, but not to other stimuli such as reward. To validate the activity-dependent labeling in TRAP2;Ai14 mice, we labeled SANs in two cohorts of mice exposed either to the home cage condition or to foot shock. Several days after

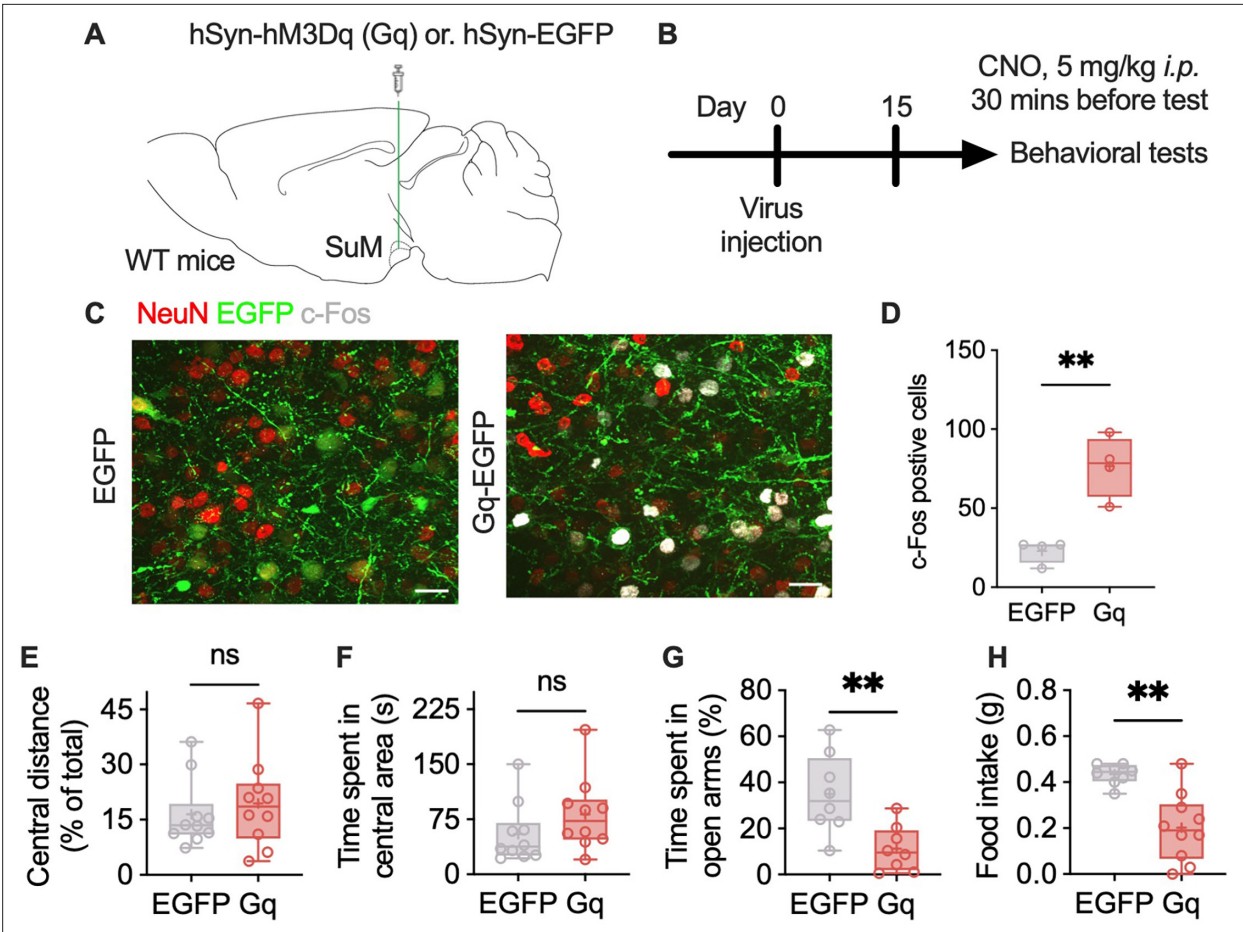

**Figure 2.** Activation of the supramammillary nucleus (SuM) produces anxiety-like behavior. (**A–B**) Virus injection information (**A**) and workflow for chemogenetic manipulation (**B**). (**C**) Representative c-Fos images. (**D**) Statistical analysis of the number of c-Fos-positive cells displayed in panel C. n=4 per group; unpaired t test. (**E–F**) Statistical analysis of the distance traveled in the central area (**E**) and the time that the mice spent in the central area (**F**) in the OF test. n=10 per group; unpaired t test for the data in (**E**) and the Mann-Whitney test for the data in (**F**). (**G**) Statistical analysis of the time that the mice spent in the open arms of the elevated zero maze (EZM). n=8 per group; unpaired t test. (**H**) Statistical analysis of sucrose pellets consumed. n=8–10 per group; Mann-Whitney test. The bars in D–H indicate the Min to Max of all data points, and the '+' indicates mean value of all data points. 'ns', p>0.05; '**', p<0.01.

The online version of this article includes the following source data and figure supplement(s) for figure 2:

**Source data 1.** The number of c-Fos+ cells in panel D, and behavioral test data in panel E-H, and corresponding statistical results.

**Figure supplement 1.** The chemogenetic manipulation of supramammillary nucleus (SuM) and stress-activated neurons (SANs) has effects on the performance of mice in the open field (OF) and the elevated zero maze (EZM).

labeling, the mice were exposed to either sucrose pellets or social stress to induce reward-related or stress-related c-Fos expression, respectively. The reactivation of SANs under reward stimulation and stress was then compared (*Figure 3D–H*). Foot shocks dramatically activated and labeled neurons in the SuM (*Figure 3F*). Social stress but not reward (presentation of sucrose pellets) induced neuronal activation (*Figure 3G*) and led to a much greater chance of reactivation of SANs (*Figure 3H*). These data suggest the specific regulatory effect of the SuM on the effects of stress but not reward.

## Reactivation of SuM^SANs promotes anxiety-like behavior

To make sure mice are on similar basal conditions while applying chemogenetic manipulation, we subjected mice to an acute stress protocol involving foot shocks and then performed the elevated plus maze (EPM) and EZM tests to evaluate anxiety on days 2 and 7 (*Figure 4A*). The mice that experienced foot shocks showed decreases in the exploration time in the open arms on day 2. However, acute stress-induced anxiety was not detected on day 7 (*Figure 4B*), which allowed us to compare

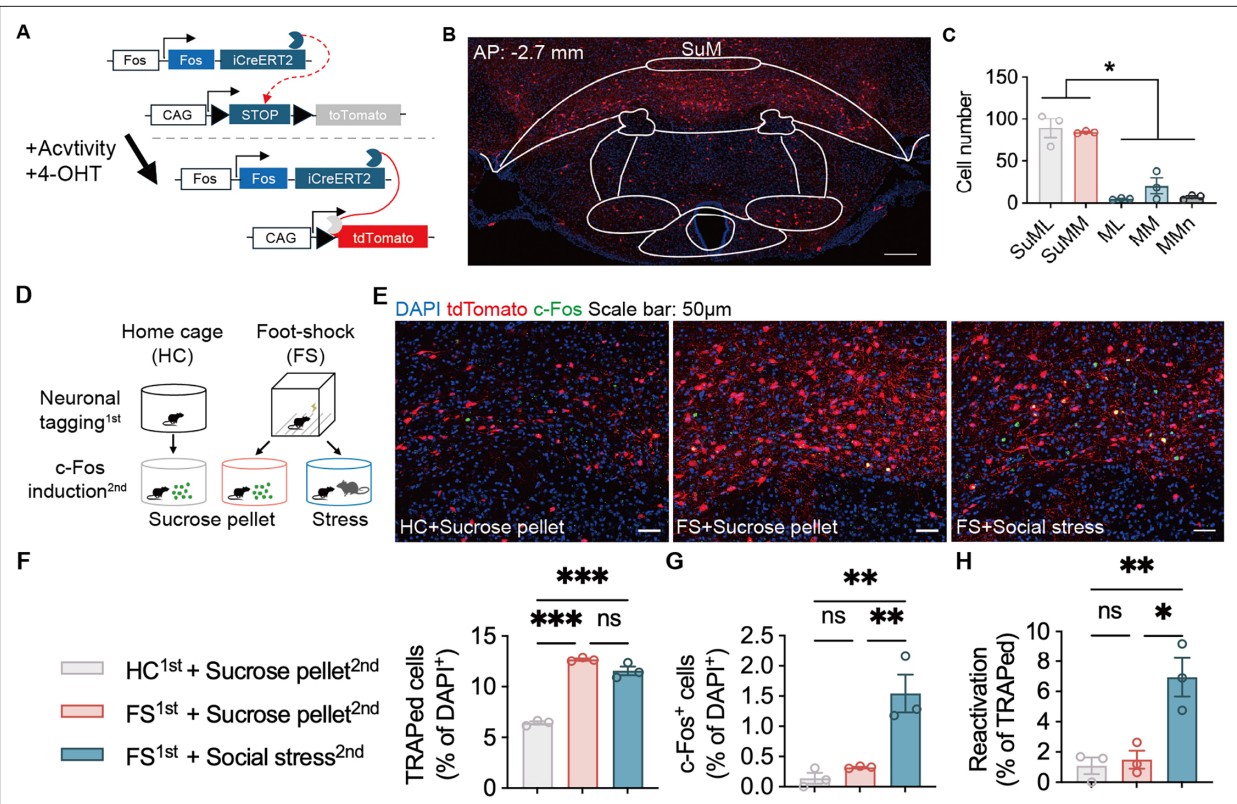

**Figure 3.** Stress-activated neurons (SANs) in the supramammillary nucleus (SuM) selectively respond to social stress but not reward. (**A**) Workflow of neuronal tagging. (**B**) Representative image of SANs in the SuM (DAPI: blue, tagged cells: red). (**C**) Quantitative statistics of stress-tagged cells in several brain areas. n=3 per area; one-way ANOVA followed by Tukey's post hoc test. (**D**) Workflow of neuronal tagging and c-Fos staining. (**E**) Representative images of stress-tagged cells and c-Fos expression induced by sucrose and social stress (DAPI: blue, tdTomato: red, c-Fos: green). (**F**) Statistical analysis of the number of stress-tagged cells in the SuM. n=3 per group; one-way ANOVA followed by Tukey's post hoc test. (**G**) Statistical analysis of the number of c-Fos+ cells after sucrose or social stress exposure. n=3 per group; one-way ANOVA followed by Tukey's post hoc test. (**H**) Statistical analysis of the reactivation of stress-tagged cells. n=3 per group; one-way ANOVA followed by Tukey's post hoc test. The data in C and F–H are presented as the means ± SEMs. 'ns', p>0.05; '*', p<0.05; '**', p<0.01; '***', p<0.001.

The online version of this article includes the following source data for figure 3:

**Source data 1.** The number of TRAPed cells in panel C and panel F, c-Fos+ cells in panel G and the reactivation ratio in panel H, and corresponding statistical results.

the reactivation of SANs produced anxiety-like behavior between groups at the same baseline. Seven days after SANs tagging, specific activation of SANs significantly increased the concentration of corticosterone (a peripheral indicator of stress) in the mouse serum (*Figure 4C and D*). Experiments involving chemogenetic manipulation also revealed the sound-selective activation of SANs in the SuM (*Figure 4E and F*). We then tested whether manipulating SANs in the SuM influences the anxiety-like behavior of the mice. The mice were subjected to the OF and EZM tests at least 1 week after SANs were tagged, followed by reward-seeking tests (*Figure 4G and H*). CNO was administered intraperitoneally 30 min before the test. Chemogenetic activation of SANs decreased the total distance traveled by the mice in the OF and EZM tests (*Figure 2—figure supplement 1D and E*). The mice also presented decreases in the distance traveled in the central area (*Figure 4I*) and time spent in the central area in the OF test (*Figure 4J*), time spent in the open arms in the EZM test (*Figure 4K*) and food consumption (*Figure 4L*). These data suggest that SANs in the SuM encode anxiety-like behavior.

## vSub-SuM projections encode anxiety-like behavior

The SuM receives afferents from various brain areas, and we identified projections to the SuM by using a nonvirus- and virus-based retrograde tracing strategy (*Figure 5A*, *Figure 5—figure supplement*

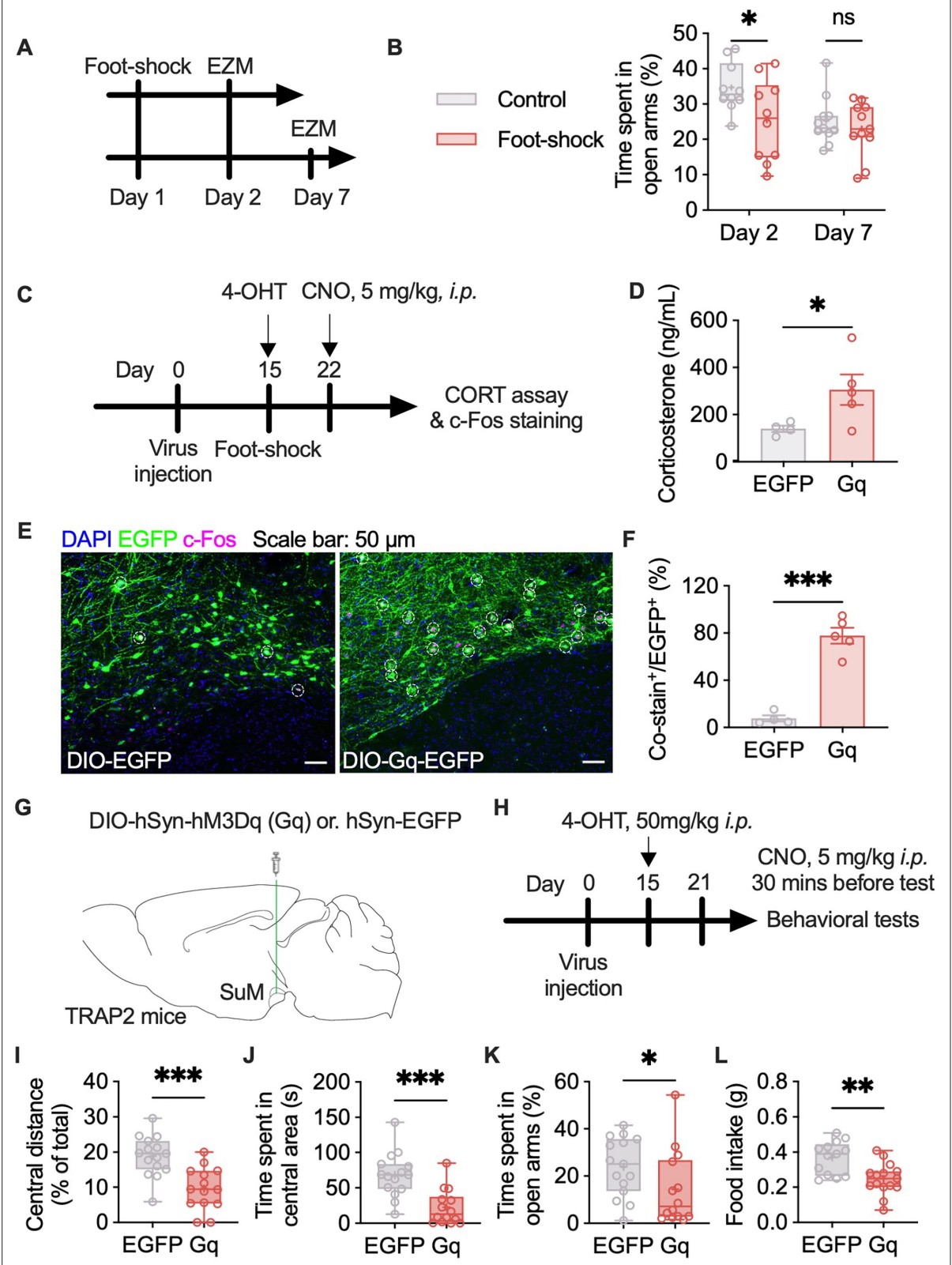

**Figure 4.** Selective chemogenetic activation of stress-activated neurons (SANs) elevates corticosterone level and produces anxiety-like behavior. (**A**) Workflow of the acute stress and anxiety tests. (**B**) Statistical analysis of the time that the mice spent in the open arms of the elevated zero maze (EZM). n=10–11 per group; two-way ANOVA followed by Sidak's post hoc test. (**C**) Workflow of the CORT assay and c-Fos staining. (**D**) Statistical analysis of the serum concentration of corticosterone after the application of clozapine N-oxide (CNO). n=4–5 per group; unpaired t test. (**E**) Representative images

*Figure 4 continued on next page*

*Figure 4 continued*

of stress-tagged cells and c-Fos expression induced by chemogenetic manipulation (DAPI: blue, EGFP: green, c-Fos: violet). (**F**) Statistical analysis of the percentage of costained cells relative to EGPF⁺ cells in the supramammillary nucleus (SuM). n=4–5 per group; unpaired t test. (**G–H**) Virus injection information and workflow of chemogenetic manipulation. (**I–J**) Statistical analysis of the distance traveled in the central area (**I**) and the time that the mice spent in the central area (**J**) in the open field (OF) test. n=14–15 per group; unpaired t test. (**K**) Statistical analysis of the time that the mice spent in the open arms of the EZM. n=14–15 per group; Mann-Whitney test. (**L**) Statistical analysis of sucrose pellets consumed. n=13–15 per group; unpaired t test. The data in D and F are presented as the means ± SEMs. The bars in B and I–L indicate the Min to Max of all data points, and the '+' indicates mean value of all data points. '*', p<0.05; '**', p<0.01; '***', p<0.001.

The online version of this article includes the following source data for figure 4:

**Source data 1.** The behavioral test data in panel B and I-L, corticosterone level in panel D, co-stain ratio in panel F and corresponding statistical results.

1A–C). Afferents from the dSub and vSub were identified using CTB-647 and adeno-associated virus (AAV) (*Figure 5—figure supplement 2A and B*). These projection neurons expressed Vglut1 RNA but not Vgat RNA (*Figure 5B*), suggesting that Sub-SuM projections are excitatory neuronal projections, as Vglut1 is a crucial marker of glutamatergic neurons. We then performed an electrophysiological experiment. Optogenetics-evoked postsynaptic currents (PSCs) in SuM neurons were blocked by perfusion with DNQX, which indicates the existence of glutamatergic projections from the Sub to the SuM (*Figure 5C–E*). To investigate how Sub-SuM projections modulate stress and anxiety-like behavior, we then used fiber photometry to measure the calcium concentration to assess the activity patterns of the projection neurons (*Figure 5F–H*). The projection neurons in the vSub, but not those in the dSub, were more strongly activated when the mice moved into the open arms from the closed arms of the EZM (*Figure 5I–M*). On the other hand, vSub, but not dSub, projection neurons show lower calcium activity while mice back to the closed arms (*Figure 5—figure supplement 3A–D*). Following exposure to acute stress, both dSub-SuM and vSub-SuM projection neurons presented increased calcium activity (*Figure 5N–R*). These data suggest that vSub-SuM projections, but not dSub-SuM projections, may participate in regulating anxiety-like behavior.

## Chronic inhibition of vSub-SuM projections alleviates anxiety-like behavior

After confirming the regulatory role of vSub-SuM projections in anxiety, we hypothesized that inhibition of this projection would alleviate chronic stress-induced anxiety. In the following experiments, the activity of these projections was chronically inhibited via a chemogenetic strategy. The mice were exposed to CSDS after the expression of the Gi protein was induced specifically on vSub-SuM projection neurons and their axons (*Figure 6A, B, and D*). The body weights of the mice were monitored throughout the entire procedure to assess their health (*Figure 6C*). The mice showed no change in social interaction test scores after CSDS exposure (*Figure 6E*). In the EZM test, the mice in which vSub-SuM projections were inhibited presented less anxiety-like behavior, as indicated by a longer time spent in the open arms of the EZM but no significant change in the distance traveled (*Figure 6F and G*). Taken together, these data suggest that vSub-SuM projections are the essential neuronal projections for regulating chronic stress-induced anxiety-like behavior.

## Discussion

In this study, we combined multiple methods to determine whether SuM is a brain region that is involved in modulating anxiety. SuM neurons strongly respond to acute and chronic stress, and their activation results in robust increases in anxiety-like behavior in mice. We then defined a small ensemble of neurons that are activated by stress, called SANs. These SANs specifically respond to stressful stimuli but not reward. Selective activation of SANs in the SuM increases the serum concentration of corticosterone and anxiety-like behavior in mice. The neuronal circuits that may underlie the regulation of anxiety were also determined in this study. The subiculum sends glutamatergic projections to the SuM and can be activated by stress, whereas only the vSub has a potential effect on the transition to anxiety. We finally determined that inhibition of SANs in the vSub project to the SuM is sufficient to alleviate anxiety in mice after CSDS exposure.

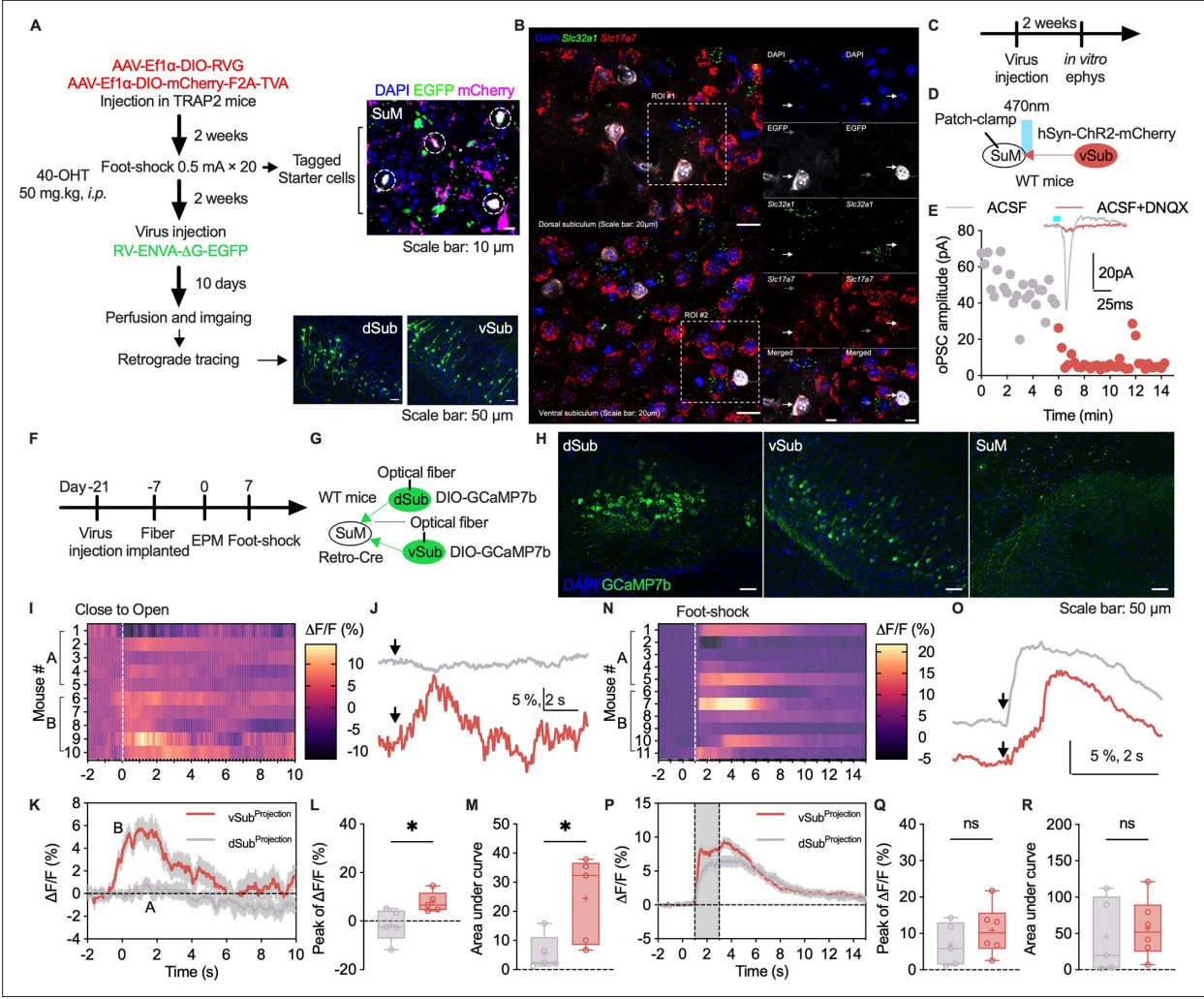

**Figure 5.** Ventral subiculum (vSub)-supramammillary nucleus (SuM) projections encoding anxiety-like behavior. (**A**) Workflow of virus-based retrograde neuronal tracing. (**B**) Representative images of in situ RNA staining (DAPI: blue, Slc32a1: green, Slc17a7: red, EGFP: white). (**C**) Workflow of ex vivo electrophysiological recording. (**D**) Schematic of optically induced postsynaptic currents (oPSCs) in the SuM. (**E**) Representative traces of oPSCs. (**F**) Workflow of Ca²⁺ imaging. (**G**) Schematic of Ca²⁺ imaging of dorsal subiculum (dSub) and vSub projection neurons. (**H**) Representative images of GCaMP7b expression in the dSub, vSub, and SuM (DAPI: blue, GCaMP7b: green). (**I**) Heatmap of the Ca²⁺ fluorescence intensity during the transition from the closed to the open arms. (**J**) Representative Ca²⁺ activity during the transition from the closed arms to the open arms. (**K**) Average ΔF/F of Ca²⁺ recorded in the dSub and vSub. (**L**) Statistical analysis of the peak Ca²⁺ activity. n=5 per group; unpaired t test. (**M**) Statistical analysis of the area under the curve of Ca²⁺ activity. n=5 per group; unpaired t test. (**N**) Heatmap of the Ca²⁺ fluorescence intensity during exposure to foot shocks. (**O**) Representative Ca²⁺ activity during exposure to foot shocks. (**P**) Average ΔF/F of Ca²⁺ recorded in the dSub and vSub. (**Q**) Statistical analysis of the peak Ca²⁺ activity. n=5–6 per group; unpaired t test. (**R**) Statistical analysis of the area under the curve of Ca²⁺ activity. n=5–6 per group; unpaired t test. The bars in L–M and Q–R indicate the Min to Max of all data points, and the '+' indicates mean value of all data points. 'ns', p>0.05; '*', p<0.05.

The online version of this article includes the following source data and figure supplement(s) for figure 5:

**Source data 1.** The Ca²⁺ peak of ΔF/F in panel L and Q, the area under curve of the Ca²⁺ trace in panel M and R, and corresponding statistical results.

**Figure supplement 1.** Specific retrograde neuronal tracing of the upstream of the stress-activated neuron (SAN) in the supramammillary nucleus (SuM).

**Figure supplement 2.** Non-virus and virus-based retrograde neuronal tracing of the upstream of the supramammillary nucleus (SuM).

**Figure supplement 3.** Calcium fiber photometry during elevated plus maze (EPM) test.

## Regulation of anxiety avoidance by the SuM

The SuM has been demonstrated to respond to novel environments, social stimulation (*Li et al., 2022b*; *Beck and Fibiger, 1995*), and stress (*Tonegawa et al., 2018*; *Sun et al., 2020*). Given these findings, we assume that the SuM may be activated by foot shocks, a quantifiable acute stressor used in animal studies. Consistent with this hypothesis, we found that the SuM robustly positively responds

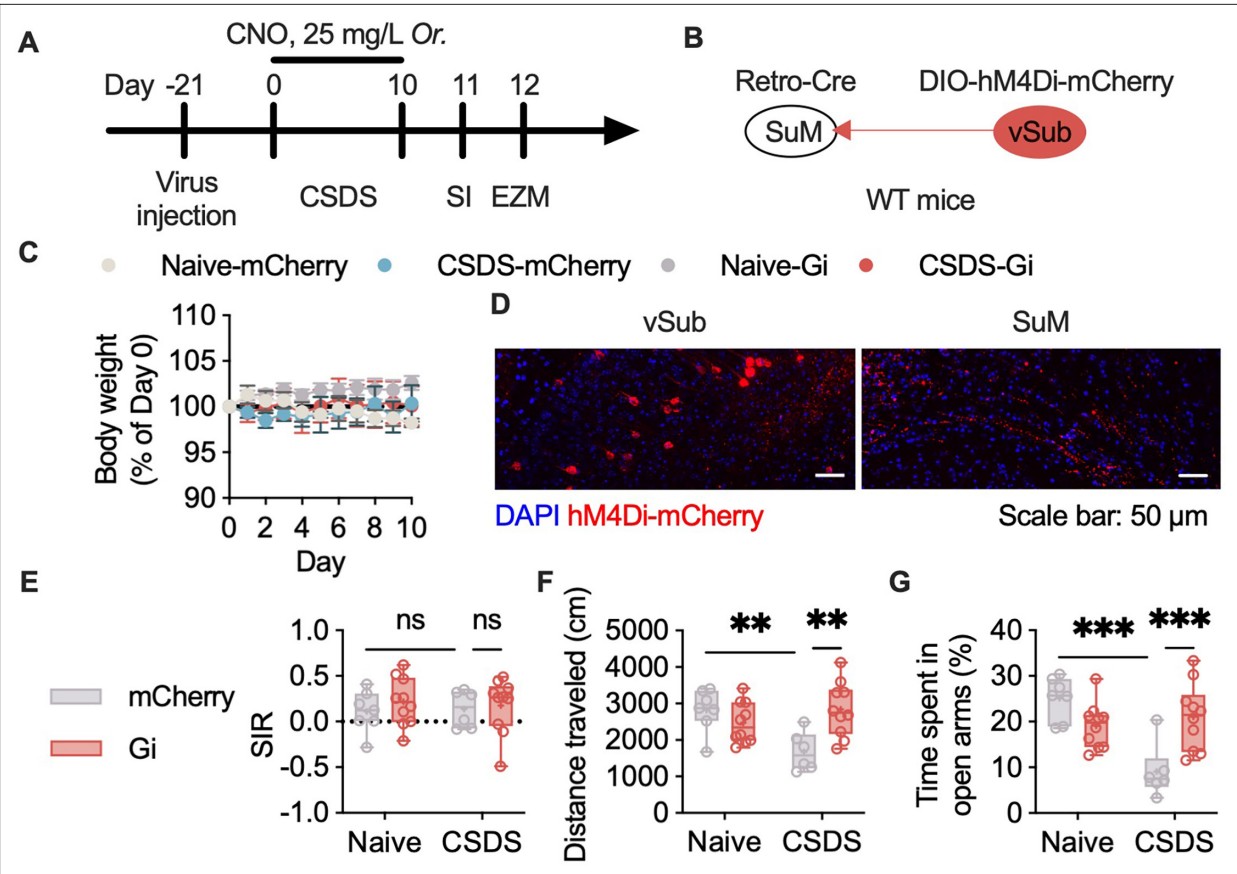

**Figure 6.** Selective inhibition of ventral subiculum (vSub)-supramammillary nucleus (SuM) projections alleviated CSDS-induced anxiety. (**A**) Workflow of CSDS and chemogenetic manipulation. (**B**) Schematic of chemogenetic manipulation of specific projections. (**C**) Body weight during CSDS exposure. (**D**) Representative images of virus expression. (**E**) Statistical analysis of the social interaction ratio after CSDS exposure. n=6–10 per group; two-way ANOVA followed by Sidak's post hoc test. (**F**) Statistical analysis of the distance that the mice traveled in the elevated zero maze (EZM). n=6–10 per group; two-way ANOVA followed by Sidak's post hoc test. (**G**) Statistical analysis of the time that the mice spent in the open arms of the EZM; two-way ANOVA followed by Sidak post hoc test. The bars in E–G indicate the Min to Max of all data points, and the '+' indicates mean value of all data points. 'ns', p>0.05; '*', p<0.05; '**', p<0.01; '***', p<0.001. CSDS: chronic social stress.

The online version of this article includes the following source data for figure 6:

**Source data 1.** The body weight in panel C, and behavioral test data in panel E-G, and corresponding statistical results.

to both acute and chronic stress through observable increases in c-Fos expression and increases in the neuronal firing rate. The activation of the SuM was also demonstrated to be essential for maintaining arousal (*Qin et al., 2022*; *López-Ferreras et al., 2019*). Sensitization to stressful events and high arousal are often associated with anxiety (*Lacagnina et al., 2019*). Thus, our data strongly suggests that the SuM potentially modulates anxiety. To further confirm whether the SuM participates in anxiety regulation, we recorded neuronal action potentials via multichannel extracellular recording while the mice were moving in the EPM, a traditional type of maze used to test anxiety in rodents. The change in the neuronal firing frequency when mice transitioned from the open arms to the closed arms supports the idea that the SuM may somewhat modulate anxiety. We then manipulated neuronal activity in the SuM via a chemogenetic method and subjected the mice to the EZM test, an improved test for assessing anxiety in rodents. The decrease in exploration time in the open arms by mice in which the SuM was activated by hM3Dq indicated increased anxiety. We noted that these results are inconsistent with those of some previous reports. Some studies have reported that lesions in the SuM and adjacent areas decrease anxiogenic behavior in rats (*Josselyn and Tonegawa, 2020*; *Zheng et al., 2024*; *Zhang et al., 2019*). López-Ferreras et al. performed the OF test, a complicated test, and reported that the chemogenetic activation of neurons in the SuM results in increased anxiety-like behaviors in

rats (*Liu et al., 2012*; *Guenthner et al., 2013*). However, further experiments involving specific tests (e.g. the EZM test) are needed to confirm whether there is a potential difference across species.

## The role of SANs in regulating anxiety avoidance

Recent studies have highlighted the importance of activity-tagged neuronal ensembles in regulating various behaviors, particularly memory (*Koren et al., 2021*; *Báez et al., 1996*; *Chen et al., 2024*; *Handa et al., 1994*), food consumption (*Yan et al., 2022*), the inflammatory response (*Forro et al., 2022*), and emotion (*Fanselow and Dong, 2010*; *Dos Santos Corrêa et al., 2019*). A negative experience-related neuronal ensemble in the hippocampus was found to increase susceptibility to chronic stress (*Fanselow and Dong, 2010*). A recent study reported that the lateral habenula contains a small population of neurons that are recruited in response to stress and mediate the development of depression in mice (*Dos Santos Corrêa et al., 2019*). These studies suggest that SANs may be important for emotional regulation. In this study, we found that the SuM was more strongly activated by acute stress than were adjacent areas. SANs were more likely to be reactivated by social stress than by sucrose reward, indicating their potential to specifically encode anxiety. The serum corticosterone concentration can be used as a marker of stress-induced change in the peripheral blood. Previous studies showed serum corticosterone can be increased by various stress stimulation (*Báez et al., 1996*; *Chen et al., 2024*; *Handa et al., 1994*; *Dos Santos Corrêa et al., 2019*); meanwhile, intentionally supplementing the diet with corticosterone can induce anxiety-like behaviors in rodents (*Peng et al., 2021*). Our data showed that the chemogenetic activation of SANs in the SuM increased the serum corticosterone concentration, whereas the inactivation of SANs had no effect, suggesting the chemogenetic manipulation of SANs may cause similar anxiety effects like real stressors. These findings, in combination with the results of the OF and EZM, suggest that SANs in the SuM are more likely involved in modulating anxiety-like avoidance. However, the reactivation rate of SANs caused by different stressors was relatively lower than the initial activation rate caused by foot shock (*Figure 3*). This suggests that stress-activated neuronal clusters may have more flexible recruitment principles, with only a small number of neurons potentially encoding emotional information, while most other neurons remain involved in encoding other neural activities. Studies in other fields, particularly studies of memory engram, have shown that the sets of neurons activated during learning are dynamic and exhibit high flexibility (*Zaki and Cai, 2024*; *Sweis et al., 2021*). While the activation of SANs produced anxiety-like behavior, the future study will examine whether silencing SuM SANs, either during stress exposure or during anxiety testing, can prevent or reduce stress-induced anxiety. We also found that both the nonselective activation of SuM neurons and the selective activation of SANs in the SuM significantly suppressed the consumption of sucrose pellets. This result may be attributed to the anxiety-induced suppression of reward seeking (*Peng et al., 2021*). However, further experiments are still needed to confirm whether this effect is anxiety dependent and whether basal food consumption is affected.

## A relevant neural circuit that regulates anxiety avoidance

The SuM recruits and is targeted by neuronal projections in the hippocampus, medial septum, and cortex (*Aranda et al., 2006*). To further understand the circuitry through which the SuM regulates anxiety, we identified projections from the dSub and the vSub to the SuM (*Tang et al., 2016*). Fiber photometry was used to measure the $Ca^{2+}$ concentration in projection neurons in the dSub and vSub, and the results revealed increased $Ca^{2+}$ activity in vSub-SuM projection neurons but not dSub-SuM projection neurons when the mice transited from the closed arms to the open arms, indicating that vSub-SuM projections encode anxiety. To confirm the regulation of anxiety by vSub-SuM projections, we exposed mice to CSDS and found that constant inhibition of vSub-SuM activity significantly abolished CSDS-induced anxiety in mice. Unlike the dorsal hippocampus, which is involved in the regulation of cognition, the ventral hippocampus is often involved in regulating emotion (*Shi et al., 2023*). The ventral CA1 area and its projections to the lateral hypothalamic area were found to mediate innate anxiety, and its activation increases anxiety-like behavior in mice (*Cumbers et al., 2007*). Although very close spatially, neurons in the subiculum are somewhat different from those in the CA1 region

(*Aggleton and Christiansen, 2015*; *Ding et al., 2020*). The vSub and its downstream brain areas were found to regulate anxiety (*Mueller et al., 2004*; *Ghasemi et al., 2022*). Jing-Jing et al. reported that the vSub and its projections to the anterior hypothalamic nucleus are essential for anxiety because the inhibition of these projections decreases anxiety-like behavior (*Kesner et al., 2023*). Our data is consistent with these findings and suggests the mediating role of the vSub and its projections to different subareas in the hypothalamus.

In summary, the activation of SuM increases anxiety-like behavior. A stressful event recruits a neuronal ensemble in SuM. The activation of SANs also significantly increases anxiety-like behavior and suppresses reward seeking. SuM receives glutamatergic projections from the vSub, and inhibition of these projections can diminish CSDS-induced anxiety-like behavior. These results suggest that SuM plays an important role in regulating anxiety-like behavior, and furthermore, studies are worth performing.

# Materials and methods

## Key resources table

| Reagent type (species) or resource | Designation | Source or reference | Identifiers | Additional information |
|---|---|---|---|---|
| Strain, strain background (*Mus musculus*) | C57BL/6J | Key Laboratory of Modern Teaching Technology, Ministry of Education | RRID:IMSR_JAX:000664 | |
| Strain, strain background (*Mus musculus*) | Fos[2A-iCreERT2] | JAX | 030323, RRID:IMSR_JAX:030323 | |
| Strain, strain background (*Mus musculus*) | Rosa26-CAG-LSL-tdTomato (Ai14) | Shanghai Model Organisms Center | NM-KI-225042 | |
| Strain, strain background (*Mus musculus*) | CD-1 | Charles River | #201 | |
| Recombinant DNA reagent | AAV2/9-hSyn-hM3Dq-EGFP | BrainVTA | PT-0891 | |
| Recombinant DNA reagent | AAV2/9-hSyn-EGFP | BrainVTA | PT-1990 | |
| Recombinant DNA reagent | AAV2/9-hSyn-DIO-hM3Dq-EGFP | BrainVTA | PT-0891 | |
| Recombinant DNA reagent | AAV2/9-hSyn-DIO-EGFP | BrainVTA | PT-1103 | |
| Recombinant DNA reagent | AAV2/Retro-hSyn-Cre | Taitool | S0278 | |
| Recombinant DNA reagent | AAV2/9-hSyn-DIO-hM4Di-mCherry | Taitool | S0193 | |
| Recombinant DNA reagent | AAV2/9-hSyn-DIO-mCherry | Braincase | BC-0025 | |
| Recombinant DNA reagent | AAV2/9-hSyn-DIO-GCaMP7b | BrainVTA | PT-2892 | |
| Recombinant DNA reagent | AAV2/9-hSyn-ChR2-mCherry | BrainVTA | PT-0150 | |
| Recombinant DNA reagent | AAV2/Retro-hSyn-EGFP | Taitool | S0237 | |
| Recombinant DNA reagent | AAV2/9-Ef1α-DIO-RVG | BrainVTA | PT-0061 | |
| Recombinant DNA reagent | AAV2/9-Ef1α-DIO-mCherry-F2A-TVA | BrainVTA | PT-0207 | |
| Recombinant DNA reagent | RV-ENVA-ΔG-EGFP | BrainVTA | R01001 | |
| Antibody | Anti-c-Fos (Rabbit Monoclonal) | Cell Signaling Technology | Cat#2250, RRID:AB_2247211 | 1:500 |
| Antibody | Donkey anti-rabbit conjugated to AF647 | Jackson ImmunoResearch | Cat#706-605-148, RRID:AB_2340476 | 1:500 |
| Commercial assay or kit | RNAscope | ACDbio | Cat#323100 | |
| Chemical compound, drug | 4-Hydroxytamoxifen | Sigma-Aldrich | Cat#H6278 | |
| Chemical compound, drug | 4-Hydroxytamoxifen | Bidepharm | Cat# BD00958757 | |

*Continued on next page*

*Continued*

| Reagent type (species) or resource | Designation | Source or reference | Identifiers | Additional information |
|---|---|---|---|---|
| Chemical compound, drug | Clozapine N-oxide | Cayman | Cat#25780 | |
| Software, algorithm | SpikeInterface | SpikeInterface | RRID:SCR_021150 | |
| Other | Antifade reagent | Invitrogen | P36981 | |

## Animals

Male C57BL/6J mice aged 12–20 weeks were used. Fos[2A-iCreERT2] (TRAP2) mice were a gift from Wenting Wang (JAX, Cat. No. 030323). Rosa26-CAG-LSL-tdTomato (Ai14) mice were purchased from the Shanghai Model Organisms Center (Cat. No. NM-KI-225042). Male CD-1 mice aged 8–10 months were purchased from Charles River (Cat. No. 201). To construct TRAP2;Ai14 mice, homozygous male TRAP2 mice and homozygous female Ai14 mice were bred. Homozygous TRAP2;Ai14 mice were maintained and used in the experiments. We performed genotyping for TRAP2 and Ai14 via PCR with the following primers: TRAP2 (wild-type: 357 bp, mutant: 232 bp): wild-type forward: GTCCGGTT CCTTCTATGCAG, mutant forward: CCTTGCAAAAGTATTACATCACG, common: GAACCTTCGAGG GAAGACG; Ai14 (wild-type: 297 bp, mutant: 196 bp): wild-type forward: AAGGGAGCTGCAGTGG AGTA, wild-type reverse: CCGAAAATCTGTGGGAAGTC, mutant forward: GGCATTAAAGCAGCGT ATCC, mutant reverse: CTGTTCCTGTACGGCATGG.

The mice were housed 4–5 per cage at a constant temperature and humidity (22 ± 1°C, 30–40% RH) on a day-night cycle (lights on from 08:00-20:00) with a fixed-intensity light source. Each mouse was acclimated to the testing environment for 1–2 min. Acclimation was performed for 3 days before the behavioral experiments. The mice were fed ad libitum and euthanized with $CO_2$ after all the tests were finished. Sample size was determined according to our previous experience. All experiments and analysis described in this study were conducted double-blindly. All laboratory procedures were conducted in accordance with the Guidelines for the Care and Use of Laboratory Animals in China and the regulations of the Animal Care and Use Committee of Shaanxi Normal University. This study protocol was reviewed and approved by the Academic Committee of the Key Laboratory of Modern Teaching Technology, Ministry of Education, Shaanxi Normal University (Approval No. L20230102-01).

## Behavioral procedure

### Acute stress exposure

The mice were exposed to acute stress according to a previously reported procedure (*Marcus et al., 2020*); specifically, they were exposed to 20 foot shocks with an intensity of 0.5 mA that were randomly delivered across 10 min. The foot shocks were delivered in a fear conditioning box (Med Associates).

### Chronic stress exposure

We used a CSDS protocol to induce anxiety and depression in the mice (*Kim et al., 2017*). Each C57BL6/J mouse was housed with one CD-1 mouse, and the mice were separated by a transparent plexiglass board with several small holes. The mice were allowed to contact each other directly for 10 min every day for 10 days. Body weight was measured and recorded every day before contact.

### OF test

The OF test was carried out in a 50×50×35 cm³ arena made of white plexiglass. The mice were allowed to move freely in the arena for 10 min, and the distance the mice traveled and the time the mice spent in the central area were recorded and analyzed.

### EPM test

The EPM consisted of two open arms (30×7 cm²), two closed arms (30×7×14 cm³), and a central area (7×7 cm²). The mice were allowed to move freely in the arena for 10 min, and the time the mice spent in the open arms was recorded and analyzed.

## EZM test

The EZM was used to test whether the mice were anxious. The EZM used in this study was made of organic glass (height of 60 cm), with an inner diameter of 51.8 cm and an outer diameter of 65 cm. The closed arms of the EZM were separated by two 15-cm-high pieces of organic glass, the outer one of which was opaque. After a 15 min habituation period, the mice were placed into the EZM and allowed to move freely for 10 min. Videos were recorded and analyzed using EthoVisionXT software. The time that the mice spent in the open arms was compared between the groups to evaluate anxiety-like behavior.

## Reward seeking

On the first day, sucrose pellets were provided for habituation. The mice were then deprived of food on the second day. On the third day, the mice were placed in a new home cage without bedding and allowed to eat freely for 2 hr. The pellets were weighed to evaluate whether the experimental manipulation influenced reward seeking by the mice.

## Social interaction test

The SIT was conducted as previously described (*Kim et al., 2017*). The SIT involved two 2.5 min phases. In the first phase (no-target phase), we placed each C57BL6/J mouse in the periphery of the arena opposite the social interaction area (SIA). We allowed the animal to explore the arena freely. In the second phase (with-target phase), each C57BL6/J mouse was placed in the arena again, with a new CD-1 mouse in the SIA. The social interaction ratio (SIR) was calculated using the following formula:

$$\text{Social interaction ratio } (\text{SIR}) = \frac{\text{Time in SIA}^{\text{With-target}} - \text{Time in SIA}^{\text{No-target}}}{\text{Time in SIA}^{\text{With-target}} + \text{Time in SIA}^{\text{No-target}}}$$

## Neuronal tagging of SANs

To specifically label SANs, TRAP2 or TRAP2;Ai14 mice were intraperitoneally (i.p.) injected with 4-hydroxytamoxifen (4-OHT, 50 mg/kg) immediately after acute stress exposure or the learning phase of the CFC test. The mice were subjected to the next experiment or test after 7 days to allow Cre-dependent recombination.

4-OHT (CAS No. 68392-35-8. Sigma, Cat. No. H6278 or Bidepharm, Cat. No. BD00958757) was dissolved in DMSO at a concentration of 62.5 mg/mL and diluted with vehicle (containing 10% Tween 80 and 80% saline) on the day of neuronal tagging. The final concentration of DMSO was kept below 10% to avoid toxicity.

## Observation of the reactivation of SANs

One week after neuronal tagging, whether previously tagged SANs were reactivated in TRAP2;Ai14 mice when they were subjected to social stress was assessed. The mice in the first group underwent neuronal tagging in their home cages, and c-Fos expression was induced by sucrose pellets. The mice in the second group were subjected to neuronal tagging in response to foot shock exposure, and c-Fos expression was induced by sucrose pellets. The mice in the third group were subjected to neuronal tagging in response to foot shock exposure, and c-Fos expression was induced by social stress (one CD-1 mouse was placed in the home cage). The mice were then sacrificed 90 min after sucrose pellet feeding or social stress exposure, and c-Fos immunofluorescence staining was performed. The number of c-Fos-positive neurons was counted to determine whether reward and cross-strain social stress could activate neurons in the SuM.

## Viral vectors

An AAV vector was used to label and manipulate specific neurons or determine the calcium concentration. To manipulate the neuronal activity in the SuM, AAV2/9-hSyn-hM3Dq-EGFP (titer: 5.00E+12 GC/mL, BrainVTA, Cat. No. PT-0891) or its control vector AAV2/9-hSyn-EGFP (titer: 5.00E+12 GC/mL, BrainVTA, Cat. No. PT-1990) was injected into the SuM of mice.

To manipulate the activity of SANs in the SuM, AAV2/9-hSyn-DIO-hM3Dq-EGFP (titer: 5.00E+12 GC/mL, BrainVTA, Cat. No. PT-0891) or its control vector AAV2/9-hSyn-DIO-EGFP (titer: 5.00E+12 GC/mL, BrainVTA, Cat. No. PT-1103) was injected into the SuM of TRAP2 mice.

To chronically inhibit vSub-SuM circuitry activity, AAV2/Retro-hSyn-Cre (titer: 2.00E+12 GC/mL, Taitool, Cat. No. S0278) was injected into the SuM, and AAV2/9-hSyn-DIO-hM4Di-mCherry (titer: 2.00E+12 GC/mL, Taitool, Cat. No. S0193) or its control vector AAV2/9-hSyn-DIO-mCherry (titer: 2.00E+12 GC/mL, Braincase, Cat. No. BC-0025) was injected into the vSub of wild-type mice.

To determine the calcium concentration in dSub/vSub-SuM projection neurons, AAV2/Retro-hSyn-Cre (titer: 2.00E+12 GC/mL, Taitool, Cat. No. S0278) was injected into the SuM, and AAV2/9-hSyn-DIO-GCaMP7b (titer: 5.00E+12 GC/mL, BrainVTA, Cat. No. PT-2892) was injected into the dSub/vSub of wild-type mice.

For the ex vivo electrophysiological experiment, AAV2/9-hSyn-ChR2-mCherry (titer: 5.00E+12 GC/mL, BrainVTA, Cat. No. PT-0150) was injected into the vSub of wild-type mice.

## Neuronal tracing

Retrograde neuronal tracing was initially performed via injection of a serotype-2 AAV vector (AAV2/Retro-hSyn-EGFP, titer: 5.00E+12 GC/mL, Taitool, Cat. No. S0237) and CTB-647 (1 μg/μL, Thermo Fisher, Cat. No. C34778) into the SuM of wild-type mice. The mice were then sacrificed after 2 weeks, and the brains were cut into coronal slices for imaging.

To precisely trace neuronal afferents projecting to SuM[SANs], AAV2/9-Ef1α-DIO-RVG (titer: 5.00E+12 GC/mL, BrainVTA, Cat. No. PT-0061) and AAV2/9-Ef1α-DIO-mCherry-F2A-TVA (titer: 5.00E+12 GC/mL, BrainVTA, Cat. No. PT-0207) were injected into the SuM of TRAP2 mice simultaneously. The rabies virus (RV) vector RV-ENVA-ΔG-EGFP (titer: 2.00E+08 IFU/mL, BrainVTA, Cat. No. R01001) was injected into the SuM 2 weeks after neuronal tagging. The mice were then sacrificed after 2 weeks, and the brains were cut into coronal slices for imaging.

## Stereotaxic surgery

The mice were anesthetized using isoflurane at a concentration of 1.5–2.0%. A virus was injected into the SuM (AP: –2.8, ML: 0, DV: –4.5 mm), dSub (AP: –2.8, ML: ±0.7, DV: –1.7 mm) or vSub (AP: –3.5, ML: ±3.0, DV: –4.6 mm) according to the experimental design. If only one type of virus needed to be injected into a single brain area, the final volume was typically 150 nL. Otherwise, the final volume of the virus mixture was 200 nL. The viruses were injected at a rate of 50 nL/min. The syringe was held in place for at least 5 min and carefully removed from the brain. The mice were then returned to their home cages, and their health was monitored on the following days. All the mice that underwent surgery were subjected to the subsequent experiment after 2 weeks or more to allow virus expression.

For fiber photometry, ceramic ferrules (outer diameter: 2.5 mm, core diameter: 0.2 mm, NA: 0.50) were inserted into the dSub (AP: –2.8, ML: ±0.7, DV: –1.5 mm) or vSub (AP: –3.5, ML: ±3.0, DV: –4.4 mm) 2 weeks after virus injection under the guidance of a laser (wavelength: 470 nm). Calcium imaging was conducted at least 1 week after ferrule implementation.

## Fiber photometry

Commercially available equipment (Thinker Tech) was used to determine the calcium concentration. The fluorescence signal was activated by a laser at 470 nm, and the signal was transmitted through a low-autofluorescence fiber-optic patch cord and rotary (doric lenses) and collected. The final activation intensity was set to ~40 μW. The sampling rate was 50 Hz for all the recordings. The mice were habituated to the fiber-optic patch cord for 3 consecutive days before recording. A TTL lasting 0.1 s was delivered by the software to mark the timepoint when the mouse moved from a closed arm to an open arm in the EPM (USB-IO box, Noldus). Continuous data were stored as *.tdms files and analyzed using custom-made software in MATLAB.

## Chemogenetic manipulation

To manipulate neuronal activity in the SuM in wild-type and TRAP2 mice, CNO (5 mg/kg; Cayman, Cat. No. 25780) was injected i.p. 30 min before behavioral tests were performed. For chronic inhibition of circuit activity, CNO was administered orally (25 mg/L).

For acute and chronic experiments, CNO was dissolved in DMSO at a concentration of 10 mg/mL and stored at –20°C or in saline at a concentration of 1 mg/mL and stored at –80°C. The storage solution was diluted with saline to a concentration of 0.75 mg/mL to prepare a working solution for acute manipulation or to a concentration of 25 mg/L to prepare a working solution for chronic inhibition on the day of the experiment.

## Immunofluorescence

The mice were anesthetized with 20% urethane and perfused with PBS or saline. The mouse brain was dissected and immersed in 4% paraformaldehyde (PFA) at 4°C overnight. The PFA solution was then replaced with a 30% sucrose solution. After the brain sank to the bottom, it was embedded in optimal cutting temperature (OCT) compound and frozen in a cryostat (CM1950, Leica). Coronal slices (40 μm) were cut and collected in a 24-well plate. After the residual OCT was removed with PBS, the slices were blocked with 0.3% Triton X-100 and 10% normal donkey serum at room temperature (RT) for 2 hr. The slices were then incubated with diluted primary antibody (rabbit anti-c-Fos, 1:500, Cell Signaling Technology, Cat. No. 2250, RRID:AB_2247211) at 4°C overnight. The next day, the slices were washed and incubated with secondary antibody dilutions (donkey anti-rabbit conjugated to AF647, 1:500, Jackson ImmunoResearch, Cat. No. 706-605-148, RRID:AB_2340476) at RT for 2 hr. After washing, the slices were transferred to slides and mounted with an antifade reagent (Thermo Fisher, Cat. No. P36981). Images of the slices were collected using a Zeiss M2 microscope and then analyzed.

## RNA fluorescence in situ hybridization

The samples were processed as described in the *Immunofluorescence* section. Slices (10 μm thick) were cut and dried at RT for ~15 min and then heated at 37°C for 30 min in a hybridization oven. The baked slides were then moved to precooled 4% PFA solution for fixation (~15 min). The slices were dehydrated in 100% ethanol at RT for 5 min. The dehydration step was then repeated. The following steps were performed as recommended by the manufacturer (ACDbio, Cat. No. 323100). To label vglut1, vglut2, and vgat RNA, the slices were hybridized with Mm-*Slc17a7* (ACDbio, Cat. No. 416631-C1), Mm-*Slc17a6* (ACDbio, Cat. No. 319171-C1) and Mm-*Slc32a1* (ACDbio, Cat. No. 319191-C3), respectively. The samples were then stained with Opal dye.

## Costaining of protein and RNA

After confirming RNA staining, the slices were blocked in 10% normal goat serum for 1 hr. The blocking solution was removed, and the slices were incubated with diluted primary antibody (mouse anti-GFP, 1:500, Thermo Fisher, Cat. No. MA5-16256; rabbit anti-tdTomato, 1:500, Oasis BioFarm, Cat. No. OB-PRB013) at 4°C overnight. The slides were washed with PBS and incubated with secondary antibody solution (goat anti-rabbit/mouse conjugated to HRP, Proteintech, Cat. No. PR30009) at RT for 1 hr (in the dark). After washing, the slices were stained with Opal dye at RT for 30 min. The slides were then mounted and imaged.

## Corticosterone assay

Mouse whole blood was collected 90 min after CNO injection (5 mg/kg, i.p.). The samples were subsequently centrifuged at 2000×$g$ for 10 min at 4°C after being left to stand at RT for 30–60 min. The supernatant was then carefully collected as the serum. Corticosterone levels were then measured using a commercial ELISA kit (Beyotime, Cat. No. PC100) according to the manufacturer's instructions.

## Ex vivo electrophysiology

The mice were anesthetized with urethane and then decapitated. The brain was quickly removed from the skull and immersed in precooled sucrose-based cutting solution (in mM, 225 sucrose, 2.5 KCl, 1.25 NaH$_2$PO$_4$, 26 NaHCO$_3$, 11 D-glucose, 5 L-ascorbic acid, 3 sodium pyruvate, 7 MgSO$_4$·7H$_2$O, 0.5 CaCl$_2$). After being fixed on a metal plate, the brain was cut into 300 μm slices. The slices were then collected and incubated in artificial cerebrospinal fluid (ACSF) containing (in mM): 122 NaCl, 2.5 KCl, 1.25 NaH$_2$PO$_4$, 26 NaHCO$_3$, 11 D-glucose, 2 MgSO$_4$·7H$_2$O, and 2 CaCl$_2$ equilibrated with 95% O$_2$-5% CO$_2$ at 28°C for at least 1 hr before recording.

To evoke PSCs using light, whole-cell recording of global SuM neurons near axons illuminated by ChR2-mCherry injected into the vSub was performed. The final light intensity at the end of the optical fiber was set to ~5 mW/mm². Then, blue light (470 nm, width: 10 ms, frequency: 0.05 Hz) was used to evoke optically induced PSCs (oPSCs). DNQX (20 µM) was perfused into the ACSF to isolate AMPA-dependent currents from the oPSCs after establishing a 5 min baseline.

## In vivo electrophysiology

The mice were anesthetized with 2% isoflurane and fixed to a stereotaxic device. A 16-channel microwire electrode array (KD-MWA, KedouBC), a 4×4 array of 25 µm NiTi wires spaced 200 µm apart, was slowly inserted into the mouse brain. Four small nails were first inserted into the skull, with a ground wire presoldered onto one of them. The electrode array was left in the SuM (AP: –2.8, ML: 0, DV: –4.55 mm), and then dental cement was used to fix it onto the skull.

The mice were introduced to the recording area at least 1 week after surgery. During the day, the electrode array attached to the mouse skull was connected to the OpenEphys acquisition board through an Intan head stage. An OpenEphys GUI was used to visualize and save electrical signals. The mice were allowed to move freely inside a home cage-like arena for at least 20 min. Only data acquired during the last 5 min were saved and then analyzed via Python-based software.

Spikes were detected and divided into single units using SpikeInterface (*Buccino et al., 2020*; https://github.com/SpikeInterface/spikeinterface; *Buccino et al., 2026*). Continuous binary raw data (sampling rate: 30 kHz) were imported and filtered using a bandpass butter filter at a cutoff value of 300 Hz. Movement artifacts were removed by subtracting medians across all channels. The templates were then extracted and fitted using SpyKING CIRCUS 2 inside the SpikeInterface frame. Neurons meeting the following criteria were excluded from the subsequent analysis: (1) spikes with refracting period violations smaller than 1 ms, accounting for more than 2% of total spikes, and (2) a total frequency lower than 0.2 Hz. Neurons with spike frequencies ≥10 Hz were considered RNs, whereas those with spike frequencies <10 Hz were considered FNs, as reported in a previous study (*Li et al., 2022b*). The local field potential was extracted and analyzed using the power spectrum analysis tool in MATLAB.

## Statistical analysis

The data are presented as the means ± SEMs in all the figures in this manuscript. For normally distributed data with equal standard deviations, independent t tests for unpaired data and dependent t tests for paired data were performed in GraphPad software to compare mean values between two groups. Otherwise, the Mann-Whitney test for unpaired data and the Wilcoxon test for paired data were performed instead. One-way ANOVA followed by Tukey's post hoc test and two-way ANOVA followed by Sidak's post hoc test were performed to compare mean values among more than three groups. A p value less than 0.05 was considered to indicate a statistically significant difference between groups. '*' represents p<0.05, '**' represents p<0.01, and '***' represents p<0.001.

## Acknowledgements

We thank Dr. Wenting Wang for his generous gift of the transgenic mice. We thank all the members of MTT for their valuable comments. This research was funded by the National Natural Science Foundation of China (No. 82371518, 82071516, and 82441060), STI 2030—Major Projects 2021ZD0200500, the Humanities and Social Science Fund of the Ministry of Education of China (No. 22XJC880005), the Innovation Capability Support Program of Shaanxi (Program No. 2021PT-055), the Natural Science Basic Research Plan in Shaanxi Province of China (Program No. 2024JC-YBQN-0902 and 2023-JC-YB-189), and the Scientific and Technological Innovation Team of Shaanxi Innovation Capability Support Plan (No. 2022TD-47).

# Additional information

## Funding

| Funder | Grant reference number | Author |
| --- | --- | --- |
| National Natural Science Foundation of China | 82371518 | Jing Han |
| National Natural Science Foundation of China | 82071516 | Zhiqiang Liu |
| National Natural Science Foundation of China | 82441060 | Jing Han |
| STI 2030-Major Projects 2021ZD0200500 | | Zongpeng Sun |
| Humanities and Social Science Fund of Ministry of Education of China | 22XJC880005 | Zongpeng Sun |
| Innovation Capability Support Program of Shaanxi | 2021PT-055 | Zhaoqiang Qian |
| Natural Science Basic Research Program of Shaanxi Province | 2024JC-YBQN-0902 | Yuan Chang |
| Natural Science Basic Research Program of Shaanxi Province | 2023-JC-YB-189 | Yanning Qiao |
| Scientific and Technological Innovation Team of Shaanxi Innovation Capability Support Plan | 2022TD-47 | Jing Han |

The funders had no role in study design, data collection and interpretation, or the decision to submit the work for publication.

## Author contributions

Jinming Zhang, Conceptualization, Data curation, Formal analysis, Investigation, Visualization, Methodology, Writing – original draft, Writing – review and editing; Kexin Yu, Investigation, Visualization, Writing – original draft; Junmin Zhang, Investigation, Visualization, Writing – review and editing; Yuan Chang, Resources, Investigation, Visualization, Writing – review and editing; Xiao Sun, Formal analysis, Investigation; Zhaoqiang Qian, Resources, Funding acquisition, Project administration; Zongpeng Sun, Zhiqiang Liu, Resources, Funding acquisition; Yanning Qiao, Resources, Investigation; Wei Ren, Resources, Supervision, Funding acquisition, Writing – review and editing; Jing Han, Conceptualization, Resources, Supervision, Funding acquisition, Visualization, Methodology, Writing – original draft, Project administration, Writing – review and editing

## Author ORCIDs

Jinming Zhang ⬡ https://orcid.org/0000-0002-4115-276X
Jing Han ⬡ https://orcid.org/0000-0001-8705-5095

## Ethics

All laboratory procedures were conducted in accordance with the Guidelines for the Care and Use of Laboratory Animals in China and the regulations of the Animal Care and Use Committee of Shaanxi Normal University. This study protocol was reviewed and approved by the Academic Committee of the Key Laboratory of Modern Teaching Technology, Ministry of Education, Shaanxi Normal University (Approval No. L20230102-01).

Reviewer #1 (Public review): https://doi.org/10.7554/eLife.108593.3.sa1
Reviewer #2 (Public review): https://doi.org/10.7554/eLife.108593.3.sa2
Reviewer #3 (Public review): https://doi.org/10.7554/eLife.108593.3.sa3

Author response https://doi.org/10.7554/eLife.108593.3.sa4

## Additional files

### Supplementary files
MDAR checklist

### Data availability
All data generated or analyzed during this study are included in the manuscript and source data files. Customized code can be accessed through GitHub (https://github.com/zjm199502/elife2026, copy archived at *Zhang, 2026*).

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
