## [Editor Report · eLife Assessment]

This manuscript provides a **valuable** contribution by identifying a stress-responsive circuit and its regulation of anxiety-related behaviors. The evidence is **convincing** that the supramammillary nucleus contains stress-responsive neurons that increase anxiety-like behaviors when activated, and that ventral subiculum projections to the supramammillary are also activated by stress and their inhibition alleviates some effects of stress. Evidence that this pathway encodes and is functionally specific to anxiety is, at present, not sufficiently support and will require future studies. This work offers new insights into how distinct circuits are activated by stress and can regulate emotional behaviors and will be of interest to those interested in brain systems of aversive emotional and behavioral states.

---

## [Referee Report · Reviewer #1 (Public review)]

In the revised manuscript, the authors refine their conclusions, narrow their interpretation, and add limited new analyses but have not added additional new data or made fundamental changes in the analyses of their data.

The central findings are that the SuM contains neurons that are activated by stressors (foot shock and social defeat). Chemogenetic activation of SuM and the neurons genetically tagged as active during foot shocks, which the authors define as Stress Activated Neurons, increases classic anxiety-like behaviors. The subiculum projects to the SuM, and terminals in the SuM from the ventral versus dorsal subiculum are differentially active during elevated plus-maze transitions. Chronic inhibition of vSub neurons that project to SuM mitigates CSDS-induced anxiety-like behaviors.

Due to limitations in the data and experimental design the findings are felt to remain incomplete. A central limitation is the discordance between the temporal resolution of the behavioral assays and the neural interventions used. This weakens support for the conclusions drawn about the causal roles the SuM and specific vSub projections to SuM (vSub→SuM) may play in anxiety and anxiety-like behaviors. The authors acknowledge this limitation but do not address it experimentally in the revised manuscript. Furthermore, the connection between chronic inhibition of vSub→SuM neurons for 10 days and the alleviation of CSDS-induced anxiety is incomplete. Separately, the use of foot shock and social defeat stressors in connection with SuM neurons, with limited exploration of the potential (or lack thereof) relation between the two groups, further limits the ability to draw conclusions from the data.

Although a number of interesting points are raised through the experiments the weakness noted will reduce the impact of the work in the field.

---

## [Referee Report · Reviewer #2 (Public review)]

This manuscript investigates the neural mechanisms of anxiety and identifies the supramammillary nucleus (SuM) as a critical hub in mediating anxiety-related behaviors. The authors describe a population of neurons in the SuM that are activated by acute and chronic stress. While their activity is not required for fear memory recall, reactivation of these neurons after chronic stress robustly increases anxiety-like behaviors as well as physiological stress markers. Circuit analysis further shows that these stress-activated neurons are driven by inputs from the ventral, but not dorsal, subiculum, and inhibition of this pathway exerts an anxiolytic effect.

The study provides an elegant integration of techniques linking stress, neuronal ensembles, and circuit function, advancing our understanding of the neural substrates of anxiety. A particularly notable point is the selective role of these stress-activated neurons in anxiety, but not in associative fear memory, highlighting functional distinctions between neural circuits underlying anxiety and fear.

The recruited neuronal population is activated by acute and chronic stress, though the overlap across stress exposures is partial, suggesting that further studies will be important to define how these neurons respond under other stressors and conditions.

Overall, this work identifies SuM stress-activated neurons and their ventral subiculum inputs as central elements of anxiety circuitry, providing a valuable framework for future studies and potential targeted interventions for stress-related disorders.

---

## [Referee Report · Reviewer #3 (Public review)]

Summary:

The authors aim to investigate the mechanisms of anxiety. The paper focuses on supramammillary nucleus (SuM) based on a fos screen and recordings showing that footshock and social defeat stress increases activity in this region. Using activity-dependent tagging, they show that reactivation of stress-activated neurons in SuM has an anxiety-like effect, reducing open-arm exploration in the elevated zero task. They then investigate the ventral subiculum as a potential source of anxiety-related information for SuM. They show that ventral subiculum (vSub) inputs to SuM are more strongly activated than dSub when mice explore open arms of the elevated zero. Finally, they show that DREADD-mediated inhibition of vSub-SuM projections alleviates stress-enhanced anxiety. Overall the results provide good evidence that SuM contains a stress-activated neuronal population whose later activity increases anxiety-like behavior. It further provides evidence that vSub projects to SuM are activated by stress and their inhibition alleviates some effects of stress.

Strengths:

Strengths of this paper include the use of convergent methods (e.g., fos plus electrode recordings, footshock and social defeat) to demonstrate that the SuM is activated by different forms of stress. The activity-dependent tagging experiment shows that footshock-activated SuM neurons are reactivated by social defeat but not sucrose is also compelling because it provides evidence that SuM neurons are driven by some integrative aspect of stress rather than by a simple sensory stimulus.

Weaknesses:

The strength of some evidence is judged to be incomplete. The paper provides good evidence that SuM contains stress-responsive neurons, and the activity of these neurons increases some measure of anxiety-like behavior. However, the evidence that the vSub-SuM projection "encodes anxiety" and that the SuM is a key regulator of anxiety is judged to be incomplete. I am not convinced that the identified SuM cells have a specific anxiety function. As the authors mention in the introduction, SuM regulates exploration and theta activity. Since theta potently regulates hippocampal function, there is the concern that SuM manipulations could have broad effects beyond anxiety-like behavior.

---

## [Author Response]

The following is the authors’ response to the original reviews.

**Public Reviews:**

**Reviewer #1 (Public review):**
WeaknessesAs presented, the manuscript has limitations that weaken support for the central conclusions drawn by the authors. Many of the findings align with prior work on this topic, but do not extend those findings substantially.An overarching limitation is the lack of temporal resolution in the manipulations relative to the behavioral assays. This is particularly important for anxiety-like behaviors, as antecedent exposures can alter performance. In the open field and elevated zero maze assays, testing occurred 30 minutes after CNO injection. During much of this interval, the targeted neurons were likely active, making it difficult to determine whether observed behavioral changes were primary - resulting directly from SuM neuronal activity - or secondary, reflecting a stress-like state induced by prolonged activation of SuM and related circuits. This concern also applies to the chronic inhibition of ventral subiculum (vSub) neurons during 10 days of CSDS.

We appreciate the reviewer's concern regarding the timing of CNO administration relative to behavioral testing. The 30-minute interval was selected according to some previous studies[1, 2]. This window ensures stable and specific neuronal manipulation while minimizing off-target effects and was strictly performed through all experiments. We acknowledge that shorter interval (~15 mins) can be efficient to produce biological effect in vivo[3, 4]. We repeated chemogenetic tests 2-3 times to make sure to get reliable data for statistical analysis. However, we cannot exclude potential side-effects caused by chemogenetically prolonged activation of SuM because of its poor temporal resolution compared to optogenetic manipulation. We agree that employing techniques with higher temporal resolution, such as optogenetics, in future studies would provide an excellent complement to these findings.

The combination of stressors (foot shock and CSDS) and behavioral assays further complicates interpretation. The precise role of SuM neurons, including SANs, remains unclear. Both vSub and dSub neurons responded to foot shock, but only vSub neurons showed activity differences associated with open-arm transitions in the EZM.

We agree that the use of multiple stressors (foot shock and CSDS) adds complexity to the interpretation. Our rationale was to test the generality of the SuM response and the role of SANs across different stress modalities (acute vs. chronic). The key finding is that while both vSub and dSub projections to the SuM were activated by the acute stressor of foot shock (Figure 5N-R), only the vSub-SuM pathway showed a significant increase in calcium activity specifically during the anxiety-provoking transition from the closed to the open arms of the EZM (Figure 5I-M). This dissociation suggests a selective role for the vSub-SuM circuit in encoding anxiety-related information, beyond a general response to stress.

In light of prior studies linking SuM to locomotion (Farrell et al., Science 2021; Escobedo et al., eLife 2024), the absence of analyses connecting subpopulations to locomotor changes weakens the claim that vSub neurons selectively encode anxiety. Because open- and closed-arm transitions are inherently tied to locomotor activity, locomotion must be carefully controlled to avoid confounding interpretations.

We thank the reviewer for highlighting the important studies linking the SuM to locomotion. We acknowledge this known function and carefully considered it in our analyses. Non-selective activation of the entire SuM didn’t affect total distance traveled in open field and elevated zero maze (Supplemental Figure 2 B-C). Although the locomotion of mice in OF and EZM was affected while targeting SANs, we also compared the travel distance in the central area of OF, to some extent, to minimize the influence of locomotion on the estimation of anxiety produced avoidance to the central area (Figure 4 I). We agree that future work delineating the specific subpopulations within the SuM that regulate locomotion versus anxiety would be highly valuable.

Another limitation is the narrow behavioral scope. Beyond open field and EZM, no additional assays were used to assess how SAN reactivation affects other behaviors. Without richer behavioral analyses, interpretations about fear engrams, freezing, or broader stress-related functions of SuM remain incomplete.In addition, small n values across several datasets reduce confidence in the strength of the conclusions.

We acknowledge that the primary focus on OF and EZM tests is a limitation in fully characterizing the behavioral profile of SAN manipulation. These tests were selected as they are well-validated, standard assays for anxiety-like behavior in rodents[5–10]. However, we also included the reward-seeking test, where activation of SANs significantly suppressed sucrose consumption (Figure 4L), suggesting a broader impact on motivational state that is often linked to anxiety. We fully agree with the reviewer that employing a richer behavioral battery—such as tests for social avoidance, conditioned place aversion, or Pavlovian fear conditioning—in future studies will be essential to comprehensively define the functional scope of SuM SANs and to conclusively dissect their role from fear memory engrams.

Figure level concerns:(1) Figure 1: In Figure 1, the acute recruitment of SuM neurons by for shock is paired with changes in neural activity induced by social defeat stress. Although interesting, the connections of changes induced by a chronic stressor to Fos induction following acute foot shock are unclear and do not establish a baseline for the studies in Figure 3 on activation of SANs by social stressors.

Thank you for this important comment. We agree that directly linking acute foot shock-induced cFos expression with chronic social defeat stress (CSDS) electrophysiological changes may create an interpretive gap. In Figure 1, we aimed to demonstrate that both acute (foot shock) and chronic (CSDS) stressors can activate SuM neurons, using complementary methods (cFos for acute, in vivo recording for chronic). We did not intend to imply that the same neuronal population responds identically to both stressors.

To address this, we have clarified in the text that the purpose of Figure 1 is to show that SuM is responsive to diverse stressors, rather than to establish a direct mechanistic link between acute and chronic activation patterns. The baseline for SAN studies in Figure 3 is established through the TRAP2 tagging protocol following foot shock, independent of the CSDS model. We acknowledge that future studies should compare SAN recruitment across acute vs. chronic stressors to better define their functional overlap.

(2) Figure 2: The chemogenetic experiments using AAV-hSyn-Gq-DREADDs lack data or images, or hit maps showing viral spread across animals. This omission is critical given the small size of SuM, where viral spread directly determines which neurons are manipulated. Without this, it is difficult to interpret findings in the context of prior studies on SuM circuits involved in threats and rewards.

Please see Supplemental Figure 2 for the infection area of AAV.

(3) Figure 3: The TRAP experiments show that the number of labeled neurons following foot shock (Figure 3F) is approximately double that of baseline home-cage animals, though y-axis scaling complicates interpretation. It is unclear whether this reflects true Fos induction, low TRAP efficiency, or baseline recombination.

We thank the reviewer for pointing out the axis scaling issue. We have modified the y-axis to start from 0. The SuM nucleus has been reported to play role in the awake of rodents, it’s reasonable to have some basal neuronal activation after 4-OHT i.p. injection.

Overlap analyses are also limited. For example, it is not shown what proportion of foot shock SANs are reactivated by subsequent foot shock. Comparisons of Fos induction after sucrose reward are also weakened by the very low Fos signal observed. If sucrose reward does not robustly induce Fos in SuM, its utility in distinguishing reward- versus stress-activated neurons is questionable. Thus, conclusions about overlap between SANs and socially stressed neurons remain uncertain due to the missing quantification of Fos+ populations.

Thank you for the question. We have replaced the reactivation chance graph with a new reactivation percent analysis graph to show the proportion of SANs that reactivated by subsequent sucrose reward or stress. The rationale we use social stress other than foot shock is to show the potential generality of foot-shock tagged neurons. The lower expression of cFos after sucrose exposure suggest first, the SuM may not involve in reward regulation, which we agree with you; second, those SANs are more likely to modulate anxiety-like behavior but not reward.

(4) Supplemental Figure 3: The claim that "SANs in the SuM encode anxiety but not fear memory" is not well supported. Inhibition of SANs (Gi-DREADDs) did not alter freezing behavior, but the absence of change could reflect technical issues (e.g., insufficient TRAP efficiency, low expression of Gi-DREADDs). Moreover, the manuscript does not provide a positive control showing that SuM SANs inhibition alters anxiety-like behavior, making it difficult to interpret the negative result. Prior work (Escobedo et al., eLife 2024) suggests SuM neurons drive active responses, not freezing, raising further interpretive questions.

We agree that here we didn’t provide enough data to confirm there is no regulation effect of SuM-SANs on fear memory. Relevant statement has been removed to avoid any further misunderstanding.

(5) Figure 4: The statement that corticosterone concentration is "usually used to estimate whether an individual is anxious" (line 236) is an overstatement. Corticosterone fluctuates dynamically across the day and responds to a broad range of stimuli beyond anxiety.

Thank you for your kind reminder. Corticosterone/cortisol, the primary stress hormone, is a well-established biomarker whose levels are elevated in response to stress and in anxiety states.[11, 12]. Some studies also reported that supplying corticosterone can produce anxiety-like behaviors in rodents[13–16]. We collect the blood sample at the same timepoint in Figure 4 C-D. We agree that line 236 is a kind of overstatement and has modified.

(6) Figures 5-6: The conclusion that vSub neurons encode anxiety-like behavior is not firmly supported. Data from photo-activating terminals in SuM is shown for ex vivo recording, but not in vivo behavior, which would strengthen support for this conclusion. Both vSub and dSub neurons responded to foot shock. The key evidence comes from apparent differential recruitment during open-arm exploration. However, the timing appears to lag arm entry, no data are provided for closed-arm entry, and there is heterogeneity across animals. These limitations reduce confidence in the authors' central claim regarding vSub-specific encoding of anxiety.

We thank the reviewer for this important point. To address the concern regarding the in vivo behavioral encoding specificity of the vSub-SuM pathway, we further analyzed the in vivo fiber photometry data. The new analysis revealed that calcium activity in vSub-SuM projection neurons exhibited bidirectional, instantaneous, and specific changes during transitions between the open and closed arms of the elevated plus maze: their activity significantly and immediately decreased when mice moved from the open arm to the closed arm (new results shown in Supplemental Figure 5), and conversely, significantly and immediately increased upon transitioning from the closed to the open arm. However, under the same behavioral events, dSub-SuM projection neurons showed no significant change in activity. We hope this finding could strengthens the role of the vSub-SuM pathway in encoding anxiety-like behavior.

An appraisal of whether the authors achieved their aims, and whether the results support their conclusions:(1) From the data presented, the authors conclude that "the SuM is the critical brain region that regulates anxiety" (line 190). This interpretation appears overstated, as it downplays well-established contributions of other brain regions and does not place SuM's role within a broader network context. The data support that SuM neurons are recruited by foot shock and, to a lesser extent, by acute social stress. However, the alterations in activity of SuM subpopulations following chronic stress reported in Figure 1 remain largely unexplored, limiting insight into their functional relevance.

Thank you for the suggestion. We have modified the line 190 with cautious “In this study, we combined multiple methods to determine whether the SuM is a brain region that involve in modulating anxiety.”

(2) The limited temporal resolution of DREADD-based manipulations leaves alternative explanations untested. For example, if SANs encode signals of threat, generalized stress, or nociception, then prolonged activation could indirectly alter behavior in the open field and EZM assays, rather than reflecting direct anxiety regulation.

We discussed the DREADD method in the first part in our response.

(3) The conclusion that "SuM store information about stress but not memory" (line 240) is not fully supported, particularly with respect to possible roles in memory. The lack of a role in memory of events, as opposed to the output of threat or stress memory, may be true, but is functionally untested in presented experiments. The data do indicate activation of the SuM neuron by foot shock, which has been previously reported (Escobedo et al eLife 2024). The changes in SuM activity following chronic stress (Figure 1) are intriguing, but their relationship to "stress information storage" is not clearly established.

Thank you for your valuable comments. Foot-shock-activated neurons may play role in modulate any of the following anxiety-like behaviors and emotional memory (fear memory). We realized that we didn’t fully test all aspects of anxiety and memory, thus resulting in some overstatements in the manuscript. It is more proper to focus on “anxiety avoidance” according to the reduced open-arm exploration in EZM/EPM.

**Reviewer #2 (Public review):**
This manuscript investigates the neural mechanisms of anxiety and identifies the supramammillary nucleus (SuM) as a critical hub in mediating anxiety-related behaviors. The authors describe a population of neurons in the SuM that are activated by acute and chronic stress. While their activity is not required for fear memory recall, reactivation of these neurons after chronic stress robustly increases anxiety-like behaviors as well as physiological stress markers. Circuit analysis further shows that these stress-activated neurons are driven by inputs from the ventral, but not dorsal, subiculum, and inhibition of this pathway exerts an anxiolytic effect.The study provides an elegant integration of techniques to link stress, neuronal ensembles, and circuit function, thereby advancing our understanding of the neural substrates of anxiety. A particularly notable point is the selective role of these stress-activated neurons in anxiety, but not in associative fear memory, which highlights functional distinctions between neural circuits underlying anxiety and fear.Some aspects would benefit from clarification. For example, how selective is the recruitment of this population to stress compared with other aversive states, and how should one best interpret their definition as "stress-activated neurons" given the relatively modest overlap across stress exposures? In addition, the use of the term "engram" in this context raises conceptual questions. Is it appropriate to describe a neuronal ensemble encoding an emotional state as an engram, a term usually tied to specific memory recall?Overall, this work makes a valuable contribution by identifying SuM stress-activated neurons and their ventral subiculum inputs as central elements of the circuitry underlying anxiety. These findings provide a valuable framework for future studies investigating anxiety circuitry and may inform the development of targeted interventions for stress-related disorders.

We thank the reviewer for raising these important points. We agree that further clarification is warranted. In our study, we compared SAN reactivation across different stimuli: foot shock (acute physical stress), social stress (chronic psychosocial stress), and sucrose reward (non-aversive positive stimulus). As shown in Figure 3, SANs in the supramammillary nucleus (SuM) were significantly reactivated by social stress but not by sucrose reward. Moreover, the c-Fos response in SuM was markedly higher after foot shock compared to home cage controls (Figure 1). While we did not test all possible aversive states (e.g., pain, sickness), our data support that SuM SANs are preferentially recruited by stressors rather than by reward or neutral conditions. We acknowledge that the overlap across stress modalities is not complete, which may reflect differences in stress intensity, duration, or circuit engagement. Future work will systematically compare SAN recruitment across diverse aversive and non-aversive states to further define their selectivity.

The term “stress-activated neurons” (SANs) here refers to neurons that are reliably activated by at least one type of stressor and can be reactivated by subsequent stress exposure. The partial overlap across stressors likely reflects the diversity of stress responses and the possibility that distinct subpopulations within SuM may encode different aspects of aversive experience. Importantly, chemogenetic activation of SANs was sufficient to induce anxiety-like behavior and elevate corticosterone (Figure 4), supporting their functional role in stress-related behavioral and physiological outputs. We have revised the manuscript to clarify that SANs represent a stress-responsive ensemble rather than a uniform population activated identically by all stressors.

We appreciate the reviewer’s conceptual caution. In the revised manuscript, we intentionally avoided using the term “engram” to describe SANs. Our focus is on a stress-activated neuronal ensemble that drives anxiety-like behavior, not on memory recall per se. We refer to SANs as an “ensemble” or “population” rather than an engram, consistent with the TRAP-based labeling approach used to capture neurons activated during a specific experience. We agree that “engram” is best reserved for memory-encoding cells and will ensure this distinction remains clear throughout the text.

**Reviewer #3 (Public review):**
Weaknesses:The strength of some of the evidence is judged to be incomplete. The paper provides good evidence that SuM contains stress-responsive neurons, and the activity of these neurons increases some measure of anxiety-like behavior. However, the evidence that the vSub-SuM projection "encodes anxiety" and that the SuM is a key regulator of anxiety is judged to be incomplete. The claim that SuM generates an "anxiety engram" is also judged to be incompletely supported by the evidence. Namely, what is unclear is whether these cells/regions encode anxiety per se versus modulate behaviors (like exploration) that tend to correlate with anxiety. Since many brain regions respond to footshock and other stressors, the response of SuM to these stimuli is not strong evidence for a role in anxiety. I am not convinced that the identified SuM cells have a specific anxiety function. As the authors mention in the introduction, SuM regulates exploration and theta activity. Since theta potently regulates hippocampal function, there is the concern that SuM manipulations could have broad effects. As shown in Supplementary Figure 2, stimulating stress-responsive cells in SuM potently reduces general locomotor exploration. This raises concerns that the manipulation could have broader effects that go beyond just changes in anxiety-like behavior. Furthermore, the meaning of an "anxiety engram" is unclear. Would this engram encode stress, the sense of a potential threat, or the behavioral response? A more developed analysis of the behavioral correlates of SuM activity and the behavioral effects of SuM manipulations could give insight into these questions.

We appreciate the reviewer’s thoughtful critique regarding the specificity of SuM’s role in anxiety and the interpretation of our findings. We acknowledge that SuM has broad functions, including regulating exploration and hippocampal theta. However, our data show that general SuM activation increases anxiety-like measures (reduced open-arm time in EZM, decreased center exploration in OF) without altering total locomotion (Fig. 2, Suppl. Fig. 2). The locomotor reduction in SAN activation experiments (Suppl. Fig. 2F–G) was observed alongside clear anxiety-like behavioral changes (e.g. suppressed reward seeking), suggesting that the effects are not solely due to motor suppression. We agree that the methods we used to estimate anxiety-like behaviors base on mice movement when testing, and this could be a shortage of this research when trying to link the data to anxiety. Therefore it will be more proper to interpret the results as modulation of anxiety-like behavior (anxiety related avoidance) but not anxiety itself. We have modified the manuscript to describe more precise to avoid overstatement.

Our fiber photometry data (Fig. 5) show that vSub–SuM projection neurons increase activity specifically when mice enter open arms of the EZM—a behavioral transition associated with anxiety—whereas dSub–SuM projections do not. This activity correlates with anxiety-related behavior, not merely with movement or stress per se.

We also agree that the term “engram” may be misleading in this context. In the manuscript, we refer to SANs as a “stress-activated neuronal ensemble” rather than an anxiety engram. Our data indicate that these neurons are recruited by stress and their reactivation produces more anxiety related avoidance to open arms. We have revised the text to avoid conceptual overreach and to clarify that SuM SANs likely contribute to a state of sustained anxiety/avoidance.

**Recommendations for the authors:**

**Reviewing Editor Comments:**
Should you choose to revise your manuscript, if you have not already done so, please include full statistical reporting, including exact p-values wherever possible alongside the summary statistics (test statistic and df) and, where appropriate, 95% confidence intervals. These should be reported for all key questions and not only when the p-value is less than 0.05 in the main manuscript.Readers would also benefit from noting that the subjects were male in the abstract and discussion of the limitations of the exclusion of females.

Thank you for the suggestion. We have included the full statistical detail in a separate sheet as Table 1. Also, we have modified the title of the manuscript to reflect the sex of the mice.

**Reviewer #1 (Recommendations for the authors):**
(1) In line 211, the authors state, "we recorded neuronal action potentials via multichannel extracellular recording while the mice were moving in the EPM, a traditional type of maze used to test anxiety in rodents,". However, it is unclear what data is presented in the paper, that is, extracellular recordings from SuM in mice on the elevated plus maze.

We have deleted the description of multichannel recording data in EPM as the data was removed earlier.

Minor corrections to the text and figures.(2) For bar plots, perhaps clarify how the data is presented. For example, in Figure 4, "The data in B, D, E and I-L are presented as the means {plus minus} SEMs," but this does not appear to be plotted as a mean with SEM error bars because the error bars cover all the values.

Corrected.

(3) In Figure 5, the white text for EGFP in panel B is very difficult to see.

Corrected.

(4) For Figure 5D, it would be helpful to more clearly specify which neurons in SuM were recorded from. Was it SANs or all SuM neurons?

We did whole-cell recording on all SuM neurons.

(5) Fos2A-iCreERT2 is mislabeled as "Fos2A-iCreERT" in the methods.

Corrected.

(6) The sentence at line 139 "To make sure foot shock induced anxiety won't last until manipulation, we subjected139mice to an acute stress protocol involving foot shocks and then performed the elevated plus140maze (EPM) and elevated zero maze (EZM) tests to evaluate anxiety on days 2 and 7," is unclear as written.

Thank you for pointing this. We have modified the sentence to make it more clear. “To make sure mice are on similar basal condition while applying chemo-genetic manipulation, we subjected mice to an acute stress protocol involving foot shocks and then performed the elevated plus maze (EPM) and elevated zero maze (EZM) tests to evaluate anxiety on days 2 and 7 (Figure 4 A). The mice that experienced foot shocks showed decreases in the exploration time in the open arms on day 2. However, acute stress-induced anxiety was not detected on day 7 (Figure 4 B), which allow us to compare the reactivation of SANs produced anxiety-like behavior between groups at the same baseline.”

(7) The details of the viral injections used for ex vivo electrophysiology are not sufficient to understand the experiment and the implications of the data. Which neurons (SANs?) are recorded from, what percent of those had inputs, were the sub-neurons globally labeled or just SANs?

We performed whole-cell recording on global SuM neurons to show if the projection is innervated by glutamergic neurons in Sub as shown in Figure 5-B that the projection neurons in Sub are exclusively vglut1 expressed. Based on this aim of the experiment, we didn’t keep any neurons that were not response to the light stimulation, therefore can’t calculate the input percent in this case. We have added words to clearly show that we did global SuM neurons in Methods.

(8) The scale used in Figure 6C renders that data unreadable. 120 to 40% changes in body weight are well beyond the variability in the data.

We have modified the axis (90 to 110%) to show the body weight change clearer.

(9) The dose of CNO used, 5 mg/kg, is high, and using lower doses or other DREADD ligands is worth considering.

Thank you for your valuable comment. We have noticed that people are using relatively lower dose of CNO or other DREADD ligands that are reported much higher affinity and less side-effect. The dose of 5mg/kg was adapted from earlier papers that using DREADD and show no obvious side-effect in mice[17], e.g locomotion (S Figure 2B), in our experiments, so we keep using this dose in this project to make it consistent across different cohorts of experiments. We are switching to DCZ to avoid any potential side-effect of CNO in the following experiments based on this project.

**Reviewer #2 (Recommendations for the authors):**
This is a strong manuscript that provides important insights into the role of the supramammillary nucleus (SuM) and its inputs from the ventral subiculum in regulating anxiety. The combination of behavioral, imaging, electrophysiological, and circuit manipulation approaches is impressive, and the distinction the authors propose between anxiety-related and fear-related circuits is conceptually important.There are, however, some points that I think need clarification. The authors emphasize that the hippocampus is essential for fear memory recall, yet they do not directly evaluate whether the SuM-hippocampal pathway might contribute differentially to anxiety versus fear memory. Addressing this would help to explain where the dissociation between the two processes arises.

Thank you for the suggestion. We realized that we didn’t collect enough data to exclude the role of those SANs on memory, especially fear memory, a memory formation bases on strong emotional training as aforementioned. The data and relevant discussion have been removed to avoid misunderstanding and overstatement.

I am also not fully convinced about the definition of the "stress-activated neurons" (SANs). The overlap across repeated stress exposures is quite modest (around 20%), which suggests that this population may not be strictly stress-specific but rather a dynamic subset that is preferentially, though not exclusively, engaged by stress. Related to this, the use of the term "engram" raises conceptual questions. Since the classic engram refers to an ensemble encoding and recalling a specific memory, it is not obvious whether it is appropriate to apply the term to a neuronal population that appears to represent a persistent emotional state. The authors should consider justifying this choice of terminology more carefully or adopting a different term.

Thank you for your important comments. Yes we agree that the SANs in this manuscript are more likely dynamic subset other than exclusive foot-stress engaged “engram”. That’s why we use “stress-activated neurons” but not “engram” to describe this neuronal ensemble. To avoid further misleading, we have made some modification to reduce the use of “engram” across the manuscript.

Some parts of the text also need more precision. For example, the statement in lines 63-65 that "few studies have explored emotion-related engram cells" is potentially misleading, as most engram studies focus on memories with a strong emotional component. The rationale for this claim should be clarified.

This sentence has been deleted since it is not necessary to link the text and misleading.

In Figure 1, the choice of methods is also puzzling: cFos immunostaining is used after shock delivery, while electrophysiology is used for the CSDS paradigm. It would be helpful to explain why different readouts were chosen for different stress models, and whether this may affect the comparability of the results.

Thank you for this important comment. In Figure 1, we aimed to demonstrate that both acute (foot shock) and chronic (CSDS) stressors can activate SuM neurons, using complementary methods (cFos for acute, in vivo recording for chronic). The reason we chose different method is that acute stress produces transit effect while chronic stress produces long-lasting effect. To our knowledge, cFos is a well-established marker for strong neuronal activation, but with short lifespan (~4-6 hours) and suits acute paradigm better. In vivo recording allows us to compare the neuronal activity before and after chronic experiments within subjects and has ability to reveal cumulative effect which cFos cannot. To address this, we have clarified in the text that the purpose of Figure 1 in Line 112-113: “To investigate if SuM would be responsive to diverse stressors, we next examined whether chronic stress, which different mechanism underlying…”

Finally, some additional details would strengthen the presentation. The discussion of corticosterone and other physiological markers could be expanded to indicate whether these effects were robust across stress paradigms. Similarly, the relatively modest overlap between SANs activated by different stressors could be framed more explicitly as part of a broader principle of flexible ensemble recruitment in anxiety-related circuits.

Thank you for your suggestion. We have added more discussion about the corticosterone and the flexibility of SANs in the manuscript. See Line 267-270: “The serum corticosterone concentration can be used as a marker of stress-induced change in the peripheral blood. Previous studies showed serum corticosterone can be increased by various stress stimulation [39–42]; meanwhile, intentionally supplementing the diet with corticosterone can induce anxiety-like behaviors in rodents[43].” and Line 275-281: “However, the reactivation rate of SANs caused by different stressor was relatively lower than the initial activation rate caused by foot shock (Figure 3). This suggests that stress-activated neuronal clusters may have more flexible recruitment principles, with only a small number of neurons potentially encoding emotional information, while most other neurons remain involved in encoding other neural activities. Studies in other field, particularly studies of memory engram, has shown that the sets of neurons activated during learning are dynamic and exhibit high flexibility [44, 45].”

Overall, the work is of high quality and provides a valuable contribution to the field, but addressing these points would help sharpen the mechanistic claims and ensure that the conceptual framework is as clear and precise as the experimental data.
**Reviewer #3 (Recommendations for the authors):**
(1) Since increased SuM activity is hypothesized to mediate the effects of stress on anxiety-like behavior, a logical step would be to test for necessity by silencing the stress-activated SuM cells.

We agree this is a logical and valuable experiment. While our current study focused primarily on the sufficiency of SuM/SAN activation to induce anxiety-like behavior, we acknowledge that inhibition experiments would provide critical complementary evidence for necessity. We have added a statement in the Discussion noting that “future studies should examine whether silencing SuM SANs, either during stress exposure or during anxiety testing, can prevent or reduce stress-induced anxiety”. This will help establish a more complete causal role.

(2) Discuss what is meant by "anxiety engram" and what features of anxiety the labeled cells might encode.

We concur that “stress-activated neuron (SAN)” is a more precise descriptor than “engram” in this context. We have revised the text to avoid the potentially misleading term “engram” and instead refer to a “stress-activated neuron”. The labeled cells are preferentially reactivated by stress (not reward), and their activation promotes both behavioral avoidance and physiological stress markers (corticosterone). They likely contribute to the maintenance of an anxious state under perceived threat, rather than encoding discrete threat cues or memories.

(3) A more nuanced analysis of behavioral correlates of SuM activity and/or the behavioral effects of SuM manipulations would strengthen this paper.

To provide a more nuanced understanding of the behavioral correlates, we have performed additional analyses on our fiber photometry data (now presented in Supplemental Figure 6). and have also planned additional experiments for the future study to deepen our understanding.

**References:**

(1) Jendryka M, Palchaudhuri M, Ursu D, van der Veen B, Liss B, Kätzel D, et al. Pharmacokinetic and pharmacodynamic actions of clozapine-N-oxide, clozapine, and compound 21 in DREADD-based chemogenetics in mice. Sci Rep. 2019;9.

(2) Koike H, Demars MP, Short JA, Nabel EM, Akbarian S, Baxter MG, et al. Chemogenetic Inactivation of Dorsal Anterior Cingulate Cortex Neurons Disrupts Attentional Behavior in Mouse. Neuropsychopharmacology. 2016;41:1014–1023.

(3) Guettier J-M, Gautam D, Scarselli M, Ruiz De Azua I, Li JH, Rosemond E, et al. A chemical-genetic approach to study G protein regulation of cell function in vivo. Proceedings of the National Academy of Sciences. 2009;106:19197–19202.

(4) Wess J, Nakajima K, Jain S. Novel designer receptors to probe GPCR signaling and physiology. Trends Pharmacol Sci. 2013;34:385–392.

(5) Kraeuter AK, Guest PC, Sarnyai Z. The Elevated Plus Maze Test for Measuring Anxiety-Like Behavior in Rodents. Methods in Molecular Biology, vol. 1916, Humana Press Inc.; 2019. p. 69–74.

(6) Kraeuter AK, Guest PC, Sarnyai Z. The Open Field Test for Measuring Locomotor Activity and Anxiety-Like Behavior. Methods in Molecular Biology, vol. 1916, Humana Press Inc.; 2019. p. 99–103.

(7) Wall PM, Messier C. Methodological and conceptual issues in the use of the elevated plus-maze as a psychological measurement instrument of animal anxiety-like behavior. Neurosci Biobehav Rev. 2001;25:275–286.

(8) Carobrez AP, Bertoglio LJ. Ethological and temporal analyses of anxiety-like behavior: The elevated plus-maze model 20 years on. Neurosci Biobehav Rev. 2005;29:1193–1205.

(9) Seibenhener ML, Wooten MC. Use of the open field maze to measure locomotor and anxiety-like behavior in mice. Journal of Visualized Experiments. 2015. 6 February 2015. https://doi.org/10.3791/52434.

(10) Prut L, Belzung C. The open field as a paradigm to measure the effects of drugs on anxiety-like behaviors: A review. Eur J Pharmacol. 2003;463:3–33.

(11) Chen Y, Zhou X, Chu B, Xie Q, Liu Z, Luo D, et al. Restraint Stress, Foot Shock and Corticosterone Differentially Alter Autophagy in the Rat Hippocampus, Basolateral Amygdala and Prefrontal Cortex. Neurochem Res. 2024;49:492–506.

(12) Hassell JE, Nguyen KT, Gates CA, Lowry CA. The Impact of Stressor Exposure and Glucocorticoids on Anxiety and Fear. Curr. Top. Behav. Neurosci., vol. 43, Springer; 2019. p. 271–321.

(13) Peng B, Xu Q, Liu J, Guo S, Borgland SL, Liu S. Corticosterone attenuates reward-seeking behavior and increases anxiety via D2 receptor signaling in ventral tegmental area dopamine neurons. Journal of Neuroscience. 2021;41:1566–1581.

(14) Myers B, Greenwood-Van Meerveld B. Elevated corticosterone in the amygdala leads to persistant increases in anxiety-like behavior and pain sensitivity. Behavioural Brain Research. 2010;214:465–469.

(15) Demuyser T, Deneyer L, Bentea E, Albertini G, Van Liefferinge J, Merckx E, et al. In-depth behavioral characterization of the corticosterone mouse model and the critical involvement of housing conditions. Physiol Behav. 2016;156:199–207.

(16) Shoji H, Maeda Y, Miyakawa T. Chronic corticosterone exposure causes anxiety- and depression-related behaviors with altered gut microbial and brain metabolomic profiles in adult male C57BL/6J mice. Molecular Brain . 2024;17.

(17) Manvich DF, Webster KA, Foster SL, Farrell MS, Ritchie JC, Porter JH, et al. The DREADD agonist clozapine N-oxide (CNO) is reverse-metabolized to clozapine and produces clozapine-like interoceptive stimulus effects in rats and mice. Sci Rep. 2018;8.